# The class 3 PI3K coordinates autophagy and mitochondrial lipid catabolism by controlling nuclear receptor PPARα

Anton Iershov [1,2,3], Ivan Nemazanyy[1,2,3,4], Chantal Alkhoury[1,2,3], Muriel Girard[1,2,3,5], Esther Barth[6], Nicolas Cagnard[7], Alexandra Montagner[8], Dominique Chretien[9,10], Elena I. Rugarli[6], Herve Guillou[11], Mario Pende[1,2,3] & Ganna Panasyuk [1,2,3]

The class 3 phosphoinositide 3-kinase (PI3K) is required for lysosomal degradation by autophagy and vesicular trafficking, assuring nutrient availability. Mitochondrial lipid catabolism is another energy source. Autophagy and mitochondrial metabolism are transcriptionally controlled by nutrient sensing nuclear receptors. However, the class 3 PI3K contribution to this regulation is unknown. We show that liver-specific inactivation of *Vps15*, the essential regulatory subunit of the class 3 PI3K, elicits mitochondrial depletion and failure to oxidize fatty acids. Mechanistically, transcriptional activity of Peroxisome Proliferator Activated Receptor alpha (PPARα), a nuclear receptor orchestrating lipid catabolism, is blunted in *Vps15*-deficient livers. We find PPARα repressors Histone Deacetylase 3 (Hdac3) and Nuclear receptor co-repressor 1 (NCoR1) accumulated in *Vps15*-deficient livers due to defective autophagy. Activation of PPARα or inhibition of Hdac3 restored mitochondrial biogenesis and lipid oxidation in *Vps15*-deficient hepatocytes. These findings reveal roles for the class 3 PI3K and autophagy in transcriptional coordination of mitochondrial metabolism.

[1] Institut Necker-Enfants Malades (INEM), 75014 Paris, France. [2] INSERM U1151/CNRS UMR 8253, 75014 Paris, France. [3] Université Paris Descartes, Sorbonne Paris Cité, 75006 Paris, France. [4] Platform for Metabolic Analyses, Structure Fédérative de Recherche Necker, INSERM US24/CNRS UMS 3633, 75014 Paris, France. [5] Pediatric Hepatology Unit, Hôpital Necker-Enfants Malades, Assistance Publique-Hôpitaux de Paris, Paris 75015, France. [6] Institute for Genetics, Cologne Excellence Cluster on Cellular Stress Responses in Aging-Associated Diseases (CECAD), University of Cologne, 50674 Cologne, Germany. [7] Plateforme Bio-informatique, Université Paris Descartes, Structure Fédérative de Recherche Necker, INSERM US24/CNRS UMS 3633, Paris 75015, France. [8] INSERM U1048, Université Paul Sabatier, Toulouse 31432, France. [9] INSERM UMR1141, Hôpital Robert Debré, Paris 75019, France. [10] Université Paris 7, Faculté de Médecine Denis Diderot, Paris 75019, France. [11] Toxalim, Université de Toulouse, INRA, ENVT, INP-Purpan, UPS, Toulouse 31027, France. These authors contributed equally: Anton Iershov, Ivan Nemazanyy. Correspondence and requests for materials should be addressed to G.P. (email: ganna.panasyuk@inserm.fr)

Metabolic homeostasis is achieved by coordinated synthesis and degradation of macromolecules. To this end, lysosomal degradation by autophagy and mitochondrial catabolism are essential in periods of nutrient shortage such as fasting[1,2]. Studies in mouse mutants have demonstrated the requirement of autophagy in the adaptation to fasting[3–5]. Notably, mitochondria contribute to energy-demanding autophagosome formation and autophagic flux by providing ATP and lipids, such as phosphatidylethanolamine[6–8]. On the other hand, autophagy supplies substrates for the oxidation reactions that take place in mitochondria, such as fatty acids and amino acids through lipophagy and proteolysis, respectively[2,9,10]. Autophagic clearance of defective mitochondria also assures mitochondrial quality control[11]. Consistently, the accumulation of dysfunctional swollen mitochondria is commonly found in autophagy deficient cells[3,12–15]. In fasting, both autophagy and mitochondrial activity are transcriptionally coordinated. This coordination, in liver tissue, relies on the activation of basic helix-loop-helix transcription factor EB (TFEB) and nutrient sensor transcription factors of the nuclear receptor superfamily including PPARα[16–20]. Of note, TFEB and the PPARα/PGC1α transcriptional complex play overlapping roles in transcriptional control of the autophagy related gene network (Coordinated Lysosomal Expression and Regulation (CLEAR) network)[18–21]. In addition, PPARα activates fatty acid β-oxidation (FAO) and ketogenesis, while PGC1α is known to be a master regulator of mitochondrial biogenesis. PGC1α co-activates nuclear respiratory factors (NRFs) and estrogen-related receptor (ERRs) transcription factors to promote the expression of proteins for mitochondrial DNA replication, mitochondrial RNA transcription as well as factors for mitochondrial maintenance[22]. Although previous studies in autophagy-deficient hepatic mutants of Atg5 and Fip200 have suggested that PPARα transcriptional responses in lipid catabolism might be suppressed, a mechanistic understanding of this dysfunction is lacking[10,23,24].

Within the autophagy network, the class 3 PI3K, present in all eukaryotes, plays a central role[25]. It functions as an obligate complex of a regulatory Vps15 subunit and a catalytic lipid kinase Vps34 subunit. Vps15 is a putative serine/threonine protein kinase required for Vps34 stability and activity[26]. The lipid kinase activity of Vps34 is a major source of the secondary messenger phosphatidylinositol 3-phosphate (PI3P)[27]. PI3P serves as a docking signal for proteins containing PI3P binding domains, such as FYVE or PX[28]. In the cell, PI3P is generated at phagophore membrane during autophagy initiation. It is also essential for endosomal sorting of plasma membrane proteins internalised by endocytosis, and it is required for delivery of hydrolases into the lysosome. In all these distinct processes, PI3P nucleates protein scaffolds to promote autophagic flux and vesicular trafficking towards the lysosome, thus placing the class 3 PI3K in control of fundamental nutrient acquisition pathways. To activate these distinct processes, the class 3 PI3K engages in different protein complexes[25]. To this end, the binding of Atg14-related protein (Atg14) or ultraviolet radiation resistance-associated gene protein (UVRAG) to the Vps34/Vps15 complex is mutually exclusive[29,30]. Atg14 stimulates Vps34 activity at phagophore membranes and is required for autophagy initiation in response to nutrient withdrawal, while the UVRAG-containing complex is implicated in endosome and autophagosome maturation[25,29,30].

Gene knockouts of either Vps34 or Vps15 revealed that the class 3 PI3K activity is indispensable for embryogenesis and organ function[31,32]. The implication of the class 3 PI3K in metabolic homeostasis is backed by the phenotypes of tissue-specific mutants. As we have previously reported, deletion of Vps15 in the liver results in defective glucose homeostasis due to increased insulin receptor signalling[33]. Recent work has further

demonstrated that even a partial inactivation of the class 3 PI3K in a mouse model of a heterozygous Vps34 lipid kinase knock-in expression, enhances insulin sensitivity and glucose tolerance[34]. However, unlike its role in autophagy, the mechanistic implication of the class 3 PI3K signalling in mitochondrial function is not fully understood. Similar to mutants of essential autophagy genes, decreased mitochondrial respiration was found in Vps34-null cells[34]. Yet, unlike in Atg-mutants, small-sized mitochondria were traced in Vps34-null hepatocytes suggesting a possible effect on mitochondrial biogenesis[13]. These observations suggest that the class 3 PI3K might contribute to the transcriptional control of mitochondrial metabolism, a hypothesis that has not yet been addressed experimentally. Here we show that hepatic expression of the class 3 PI3K is essential for metabolic adaptation to starvation in the liver through the control of PPARα transcriptional activity. We demonstrate that the availability of PPARα ligands and expression of co-activator PGC1α, as well as levels of PPARα repressors NCoR1 and Hdac3 are affected in mice with liver-specific inactivation of Vps15. Mechanistically, we show that Hdac3 and NCoR1 repressors interact with Autophagy-related protein 8 (Atg8)-like proteins and degrade in lysosome, a mechanism that is non-functional in autophagy deficient Vps15-null hepatocytes. Notably, the pharmacologic inhibition of histone deacetylases and administration of a PPARα synthetic ligand was sufficient to significantly improve mitochondrial function in Vps15-null livers. Finally, we show that, in addition to its known role in transcriptional control of ketogenesis and FAO, PPARα acts upstream of mitochondrial biogenesis in the liver. Ultimately, we propose that the class 3 PI3K acts upstream of nuclear receptors and exerts a broad transcriptional control in the liver to match autophagic activity with mitochondrial metabolism during fasting.

## Results

**Mitochondria are dysfunctional in Vps15-LKO mice.** In our recent work, we demonstrated that deletion of the regulatory subunit of the class 3 PI3K, Vps15, in hepatocytes provoked metabolic rearrangements that were mirrored by modified transcription of metabolic enzymes[33]. It led us to hypothesise that the class 3 PI3K exerts a broad transcriptional control in liver. To test this, we performed microarray analyses upon inactivation of the class 3 PI3K. To avoid a possible adaptation to a chronic loss of the class 3 PI3K, Vps15 was acutely depleted in livers of Vps15[f/f] mice using adenoviral vectors expressing Cre recombinase or GFP protein as a control. This acute liver-specific deletion of Vps15 resulted in significant liver hypertrophy and autophagy block[33]. Following analyses demonstrated that transcript levels of 2693 unique genes were modified more than 1.5-fold. Among those 1225 (45%) were significantly downregulated and 1471 (55%) were significantly upregulated (Supplementary Data 1). The subsequent Gene Ontology (GO) pathway analyses among the downregulated genes showed an enrichment of mitochondria, peroxisome and metabolic processes associated with those organelles; notably lipid metabolism (Fig. 1a; Supplementary Data 2). The GO analyses among the upregulated genes demonstrated that, consistent with the defective autophagy, there was an enrichment of oxidative stress response (Supplementary Fig. 1a, Supplementary Data 2). Further analyses by real-time quantitative PCR (RT-qPCR) of liver-specific Vps15-deficient AlbCre[+]; Vps15[f/f] mice[33], hereafter referred to as Vps15-LKO, confirmed these initial observations. Notably, marked oxidative stress in liver tissue of Vps15-LKO mice was evidenced by increased expression and activity of key transcription factor for antioxidant gene expression induction Nuclear factor erythroid 2-related factor 2 (Nrf2) (Supplementary Fig. 1b-d). Remarkably, among

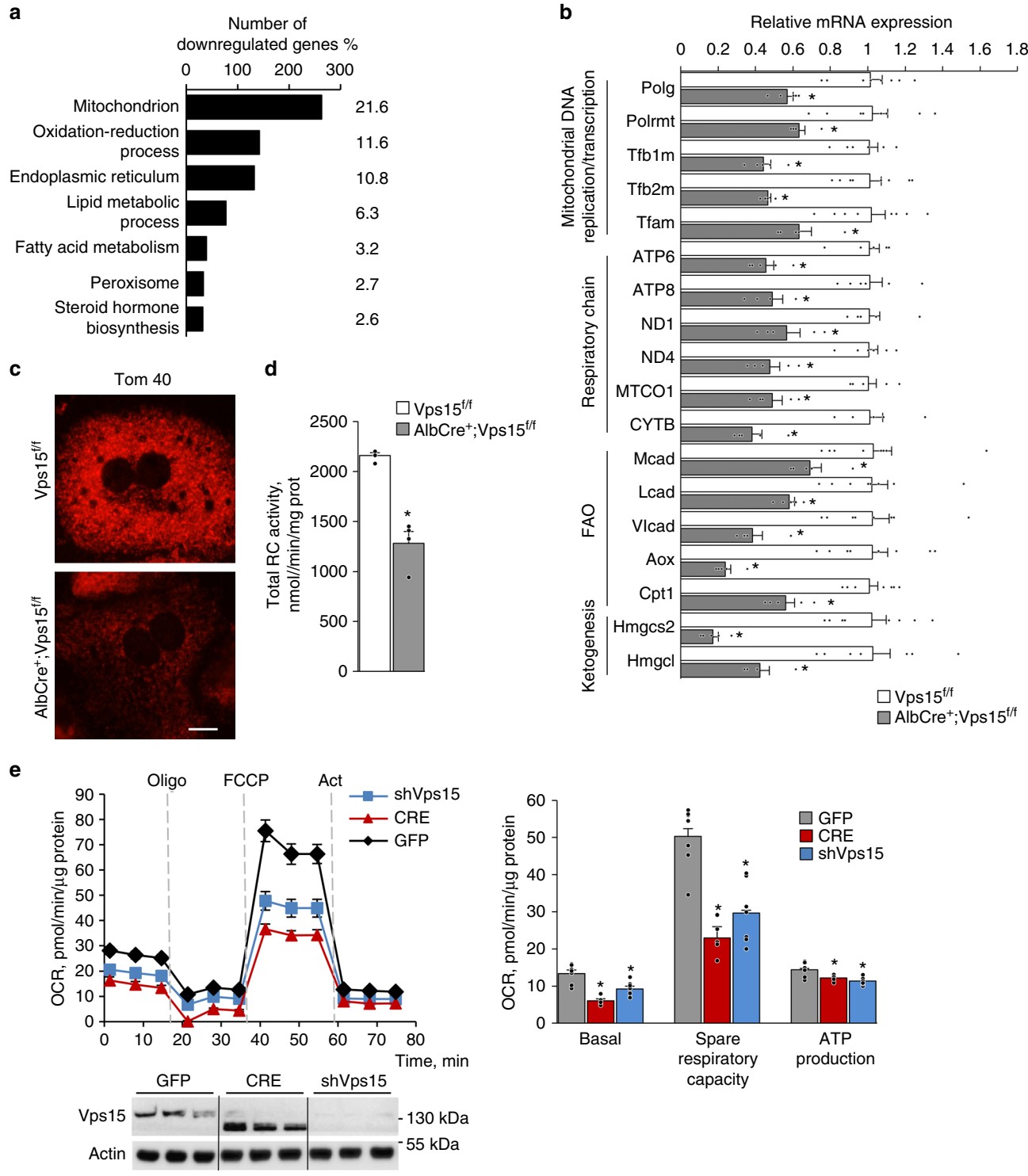

the downregulated genes, the transcript levels of enzymes involved in FAO; ketogenesis; and the expression of both nuclear and mitochondrial genome-encoded genes implicated in mitochondrial function and biogenesis were significantly decreased (by 40–80%) in liver tissues of Vps15 mutants (Fig. 1b). These defects at the transcriptional level were accompanied by mitochondria depletion as seen by decreased Tom40 and cytochrome c staining in primary hepatocytes isolated from Vps15-LKO mice and significantly decreased mtDNA content in liver tissue of Vps15-LKO mice (Fig. 1c, Supplementary Fig. 2a and 2b). Mitochondrial dysfunction was also evidenced by decreased total

respiratory chain complexes activity measured in liver extracts of Vps15-LKO mice (Fig. 1d). The later was due to a 20–60% decrease in activity of individual respiratory chain complexes (Supplementary Fig. 2c). This decreased enzymatic activity of respiratory chain complexes was accompanied by increased activity of lactate dehydrogenase in liver tissue of Vps15-LKO mice suggesting activated glycolysis (Supplementary Fig. 2d). These findings in liver tissue were also corroborated by analyses in primary hepatocytes. The metabolic profiling of mitochondria using Seahorse Bioanalyzer showed that acute depletion of Vps15 using specific shRNA or by expressing Cre recombinase in

**Fig. 1** Mitochondrial gene transcription programme is defective in Vps15-LKO mice. **a** Functional annotation clustering by Database for Annotation, Visualization and Integrated Discovery (DAVID) bioinformatic tool of significantly downregulated genes in the microarray analyses of liver tissue after acute Vps15 depletion. Percentage of genes among downregulated genes attributed to listed processes is indicated. Liver tissue was harvested ten days after transduction with adenoviral vectors expressing Cre recombinase to deplete Vps15 or GFP control protein. Significantly modified genes are listed in Supplementary Data 1. Functional annotation clustering results are listed in Supplementary Data 2. **b** Relative transcript levels of genes implicated in mitochondrial DNA replication, transcription, respiratory chain subunits and metabolic enzymes in livers of control and Vps15-LKO mice analysed by RT-qPCR. Data are means ± SEM ($n = 6$–8 for Vps15$^{f/f}$, $n = 5$ for AlbCre$^+$;Vps15$^{f/f}$, $P < 0.05$ *: vs Vps15$^{f/f}$ mice, two-tailed, unpaired Student's $t$ test). **c** Immunofluorescent analyses of Tom 40 in control and *Vps15*-null primary hepatocytes. Cells were PFA fixed and stained with anti-Tom 40 antibody, secondary anti-rabbit IgG Alexa Fluor 568 antibody was used for detection. Scale bar: 50 μm. **d** The total respiratory chain activity normalized to total protein content measured in liver tissue extracts of Vps15$^{f/f}$ and AlbCre$^+$;Vps15$^{f/f}$ mice. Data are means ± SEM ($n = 4$ for Vps15$^{f/f}$ and AlbCre$^+$;Vps15$^{f/f}$, $P < 0.05$ *: vs Vps15$^{f/f}$, two-tailed, unpaired Student's $t$ test). **e** The oxygen consumption rate measured by SeaHorse Bioanalyzer in primary Vps15$^{f/f}$ hepatocytes 48-h post-transduction with adenoviral vectors expressing GFP, Cre or shRNAVps15 under basal conditions (initial rates) and in response to sequential treatment with Oligomycin (respiration associated with ATP production), FCCP (maximal respiration), and Rotenone/Antimycin A (non-mitochondrial respiration). Dashed lines indicate the time of the addition of each reagent. Representative experiment of five independent hepatocyte cultures is presented. Quantification of basal respiration, ATP production and maximal respiratory capacity are shown on the graphs (right panel). Data are means ± SEM, $P < 0.05$ *: vs GFP-infected primary hepatocytes, two-tailed, unpaired Student's $t$ test. Lower panel shows the control immunoblot analysis of total protein extracts of hepatocytes using indicated antibodies. The immunoblot with anti-actin antibody served as a loading control

Vps15$^{f/f}$ primary hepatocytes resulted in inhibition of basal and spare mitochondrial respiration as well as decreased ATP production capacity (Fig. 1e). Of note, the lower migrating band of Vps15 is a truncated non-functional form of the protein that is transcribed from the start codon in exon 4 upon effective Cre-mediated recombination in Vps15 gene locus[31]. Importantly, the inhibition of mitochondrial respiration and metabolic gene expression in Vps15-depleted hepatocytes was not due to expression of Cre-recombinase, as these effects were not observed in transduced primary hepatocytes isolated from wild-type mice (Supplementary Fig. 2e and 2f). Therefore, deletion of *Vps15* in hepatocytes results in a transcriptional inhibition of mitochondrial maintenance program that manifests as mitochondrial dysfunction.

**Fatty acid degradation is blocked in livers of Vps15-LKO mice.** To get further insights in the metabolic rearrangements in Vps15-LKO mice, the targeted metabolomics analyses by mass spectrometry were performed on liver tissue of 6-h fasted animals. The pathway analyses of metabolomics data using MetaboAnalyst 4.0 tool suggested significant changes in lipid metabolism (Fig. 2a; Supplementary Data 3). Indeed, the long-chain-carnitine conjugates (C6-C16-carnitines) were accumulated in livers of Vps15-LKO mice (Fig. 2b). This was paralleled by decreased levels of free fatty acids and lower levels of Acetyl-CoA in livers of Vps15-LKO mice (Fig. 2b). Importantly, in agreement with defective fatty acid degradation in mitochondria, measurements of the plasmatic levels of ketone body metabolite, hydroxybutyrate, revealed a 2.7-fold lower levels in fasted Vps15-LKO mice (Fig. 2c). Altogether, metabolomics analyses advocate that Vps15 mutant hepatocytes do not sustain functional mitochondrial oxidation of fatty acids in fasting.

**PPARα is inhibited in *Vps15*-null liver.** The metabolic dysfunction of Vps15-LKO mice was reminiscent of a PPARα deficiency, whose transcriptional activity in the liver is essential for fatty acid uptake, transport, β-oxidation and ketogenesis[35–37]. Similar to reported mouse mutants of PPARα, Vps15-LKO mice were hypoketogenic (Fig. 2c) and hypoglycaemic[33,35,36]. Furthermore, bioinformatic analyses using EnrichR database of overrepresented transcription factors associated with differentially expressed genes in Vps15-null liver demonstrated proinflammatory and hematopoietic transcription factors on the top of the list for upregulated genes while RXR and PPARα as the most significantly associated with downregulated gene expression

signature (Supplementary Fig. 3a). The likely involvement of PPARα in the phenotype of Vps15-LKO mice was also corroborated by hepatic gene expression profile in liver tissue upon acute deletion of Vps15, which unveiled lipid catabolism among the statistically enriched pathways (Fig. 1a). The comparative microarray analyses between *Vps15*-null livers and liver tissue from PPARα mutants (whole-body[17] and hepatocyte-specific[37]) demonstrated a notable overlap in transcriptional responses (Fig. 2d, Supplementary Data 4). To address the status of PPARα in a condition of its maximal activation, Vps15-LKO mice were challenged by a 24-h fasting. The activation of PPARα in livers of control mice was evidenced by a induction of transcript levels of its bona fide targets in fatty acid transport to mitochondria, ketogenesis and fatty acid oxidation, *Cpt2, Hmgcs2* and *Aox*, respectively (Fig. 2e). Importantly, their transcriptional induction in response to fasting was blunted in livers of Vps15-LKO mice (Fig. 2e). In agreement with its transcriptional inhibition, nuclear PPARα protein was depleted and its transcript levels downregulated in *Vps15*-null livers (Fig. 2f and Supplementary Fig. 3b). Proteosomal degradation is established mechanism of PPARα protein turnover[38,39]. In line with lower protein levels of PPARα protein, PPARα polyubiquitination was upregulated in livers of Vps15-null mice despite its lower protein expression detected in total extracts (Supplementary Fig. 3c and 3d). This was accompanied by increased in Vps15-LKO mice levels of Huwe1, a reported E3 ubiquitin ligase for PPARα, (Supplementary Fig. 3d)[39]. Direct ligand binding controls PPARα protein expression, stability, and nuclear localization[38,40]. Fatty acids are known natural PPARα ligands[41–44]. They are delivered to the liver from adipose tissue upon fasting-induced lipolysis[37], as well as being synthesized in hepatocytes de novo[43]. As expected, the 24-h fasting potently induced steatosis in the liver of control animals, showed as the pale colour of the organ and histological findings of vacuolation in hepatocytes (Supplementary Fig. 4a and 4b). Notably, these responses were not apparent in Vps15-LKO mice as their livers were already pale and hepatocytes were markedly vacuolated (Supplementary Fig. 4a and 4b). Biochemical analyses of triglyceride levels confirmed neutral lipid accumulation in liver tissue of control mice, a response that was blunted in Vps15-LKO mice (Supplementary Fig. 4c). The dampened response to fasting in livers of Vps15-LKO mice was also evidenced by unmodified liver hypertrophy, unlike decreased liver size in fasted control mice (Supplementary Fig. 4d). To get further insight in the lipid mobilisation in Vps15-LKO mice, the biochemical analyses of plasma in fed and fasted mice were performed. Those have demonstrated that consistent with induced by fasting lipolysis,

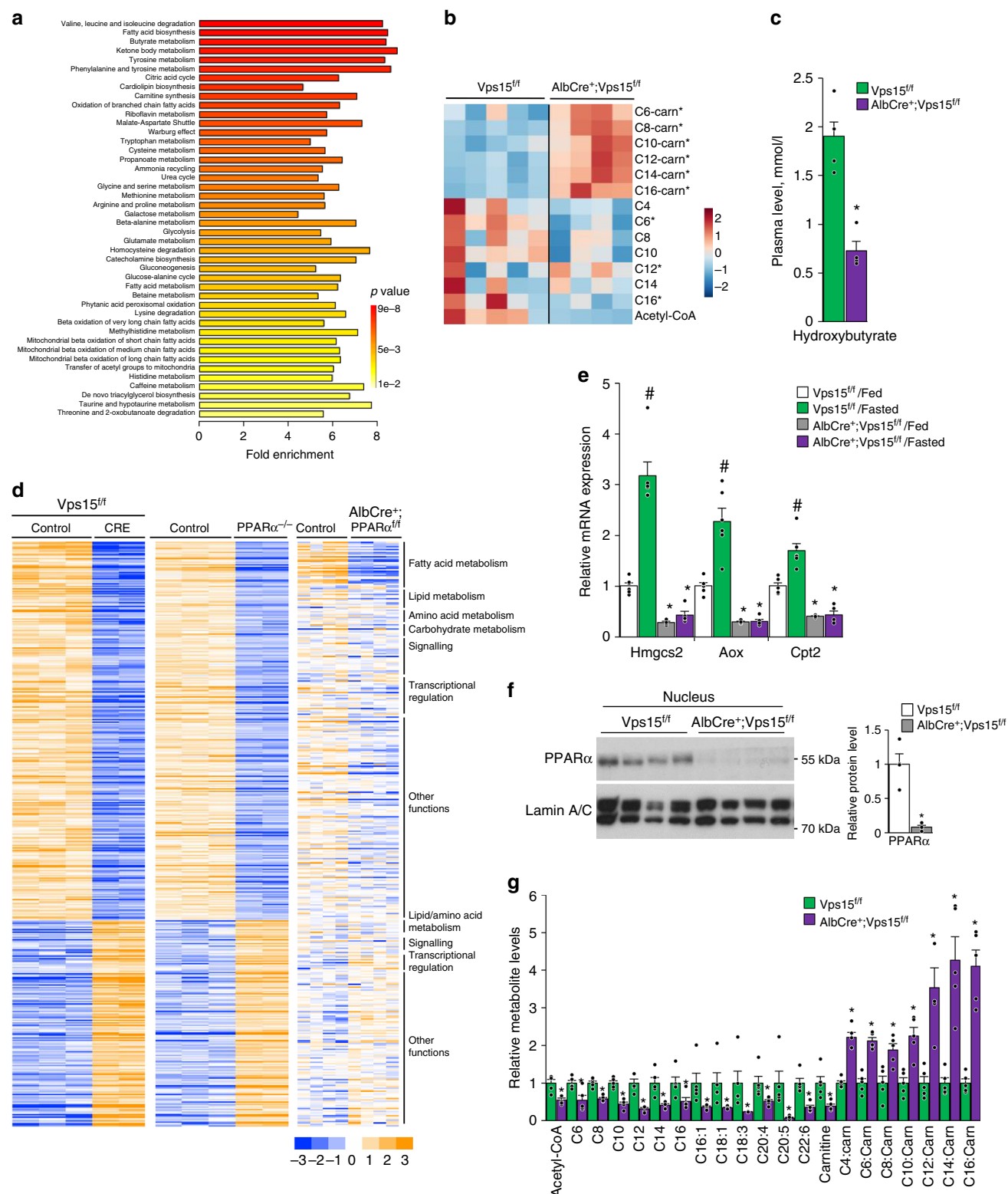

the levels of circulating glycerol and non-esterified free fatty acids were increased in plasma of control mice (Supplementary Fig. 4e). At the same time, consistent with effective uptake by liver, the plasmatic levels of triglycerides were decreased in fasted wild-type animals (Supplementary Fig. 4e). Notably, in Vps15-LKO mice, the levels of non-esterified free fatty acids and triglycerides were higher in fed state with fasting further increasing only the levels of non-esterified free fatty acids (Supplementary Fig. 4e). On the

contrary, the lactate levels were decreased albeit did not reach statistical significance in the plasma of fed Vps15-LKO mice (P value of 0.059, unpaired two-tailed Student's t-test) and fasting further lowered it in both genotypes (Supplementary Fig. 4e). Adipose tissue lipolysis is a major source of circulating fatty acids in fasting. Consistent, 24-h starvation resulted in a decrease in whole body adipose tissue content measured by dual energy X-ray absorptiometry (DEXA) scan in control and Vps15-LKO mice. In

**Fig. 2** Defective PPARα and fatty acid oxidation in liver of *Vps15* mutant. **a** Summary plot for metabolite set enrichment analysis (the colour of the bar indicates the significance from red to yellow). All identified metabolites are listed in Supplementary Data 3. Liver tissue of six week old Vps15$^{f/f}$ ($n = 5$) and AlbCre$^+$;Vps15$^{f/f}$ ($n = 4$) mice was collected for analyses after 6-h fasting. **b** Heat map showing relative levels of the fatty acids and Acyl-carnitine derivatives measured by mass spectrometry in mouse liver collected as in **a**. Each column corresponds to a different Vps15$^{f/f}$ and AlbCre$^+$;Vps15$^{f/f}$ animal and the colour of the cell indicates the relative content of the metabolite (from blue to red). $P < 0.05$ *: vs Vps15$^{f/f}$. **c** Hydroxybutyrate levels in plasma of six week old Vps15$^{f/f}$ and AlbCre$^+$;Vps15$^{f/f}$ mice that were treated as in **a**. Data are means ± SEM ($n = 5$ for Vps15$^{f/f}$, $n = 4$ for AlbCre$^+$;Vps15$^{f/f}$, $P < 0.05$ *: vs Vps15$^{f/f}$, two-tailed, unpaired Student's *t* test). **d** Heat map representing data from a gene expression profiling experiments performed with liver samples of mice of indicated genotypes (E-MTAB-7685 for Vps15, GSE35015 for PPARα$^{-/-}$, GSE73298 for AlbCre$^+$;PPARα$^{f/f}$). All genes expression of which was significantly modified were included in the analyses. The colour of the cell indicates the relative change of expression (from blue to yellow). The genes are grouped according to their biological function in GO annotation. The gene names corresponding to GO groups are listed in Supplementary Data 4. **e** Relative transcript levels of rate limiting enzymes in fatty acid transport to mitochondria, ketogenesis and FAO in the livers of 6 week old random-fed or 24-h fasted prior to sacrifice Vps15$^{f/f}$ and AlbCre$^+$;Vps15$^{f/f}$ mice. Data are means ± SEM ($n = 6$ for Vps15$^{f/f}$, $n = 4$–5 for AlbCre$^+$;Vps15$^{f/f}$, $P < 0.05$ *: vs Vps15$^{f/f}$, #: vs fed, two-tailed, unpaired Student's *t* test). **f** Immunoblot analysis of nuclear protein liver extracts of random-fed six week old Vps15$^{f/f}$ and AlbCre$^+$;Vps15$^{f/f}$ using indicated antibodies. Immunoblot with LaminA/C antibody served as a loading control. Densitometric analyses of protein levels normalised to LaminA/C levels presented as folds over Vps15$^{f/f}$-chow condition. Data are means ± SEM ($n = 4$ for Vps15$^{f/f}$, $n = 4$ for AlbCre$^+$;Vps15$^{f/f}$, $P < 0.05$ *: vs Vps15$^{f/f}$, two-tailed, unpaired Student's *t* test). **g** Relative levels of free fatty acids, carnitine and Acyl-carnitine in livers of 24-h fasted Vps15$^{f/f}$ and AlbCre$^+$; Vps15$^{f/f}$ mice measured by mass spectrometry. Data are means ± SEM ($n = 5$, $P < 0.05$ *: vs Vps15$^{f/f}$, two-tailed, unpaired Student's *t* test)

line with important dyslipidaemia, Vps15-LKO mice showed decreased adipose tissue content already in fed state which was further reduced by fasting (Supplementary Fig. 4f). This decreased adiposity in Vps15-LKO mice was paralleled by significantly increased lean mass and no changes in body fluid levels (Supplementary Fig. 4g and 4h). Moreover, despite presented with decreased adiposity, the fasting induced loss of fat was higher in Vps15-LKO as compared to wild-type mice albeit did not reach statistical significance (P value of 0.056, unpaired two-tailed Student's *t*-test) (Supplementary Fig. 4i). Finally, consistent with the lower lipid levels measured in liver of Vps15-LKO mice, the mass spectrometry analyses further confirmed a decrease in free fatty acids, including unsaturated fatty acids, in livers of fasted Vps15-LKO mice (Fig. 2g). Similar to analyses in six-hour fasted mice, acyl-carnitine derivatives were accumulated in liver tissue of 24-h fasted and in fed Vps15-LKO animals (Fig. 2g and Supplementary Fig. 4j). These analyses also showed comparable levels of free carnitine in livers of fed mice, however, those were significantly decreased in livers of fasted Vps15-LKO compared to fasted wild-type mice (Fig. 2g and Supplementary Fig. 4j). Altogether, these findings suggest that hepatic expression of Vps15 is required for whole body lipid homeostasis. Furthermore, in liver, Vps15 acts upstream of PPARα and is necessary for its expression and transcriptional activity to assure efficient fatty acid degradation during fasting.

**Fenofibrate treatment restores lipid catabolism in Vps15-LKO mice.** The lack of endogenous PPARα ligands is a likely explanation for the dysfunction of PPARα in Vps15-LKO mice. The available selective synthetic ligands of PPARα, like fenofibrate, are efficient in pharmacotherapy of hyperlipidaemias by promoting fatty acid uptake, transport and oxidation[45]. Given that residual PPARα protein was expressed in livers of Vps15-LKO mice (Fig. 2f), we hypothesized that fenofibrate treatment could be efficient in restoring PPARα transcriptional activity and correcting metabolic defects in the *Vps15* hepatic mutant. To this end, Vps15-LKO mice were treated during two weeks with fenofibrate incorporated in food. This short-term treatment normalized triglyceride levels in the plasma of Vps15-LKO mice (Supplementary Fig. 5a). Importantly, fenofibrate treated control mice induced a robust increase in transcript and protein expressions of key PPARα target genes involved in fatty acid transport and catabolism (Fig. 3a, b). These include PPARα targets for peroxisomal function (*Pex5* and *Pex6*), fatty acid transport (*Fabp1* and *Cpt1*), ketogenesis and FAO (*Hmgcs2*, *Lcad* and

*Aox*). Importantly, in Vps15-LKO mice, administration of the synthetic ligand was sufficient to restore PPARα transcriptional activity, at least to levels observed in chow-fed control mice (Fig. 3a, b). Consistent with its transcriptional activation, nuclear levels of PPARα as well as Fabp1 protein, which is required for delivery of PPARα ligands, were induced by fenofibrate treatment (Fig. 3c; Supplementary Fig. 5b). Therefore, administration of synthetic ligand restores PPARα transcriptional activity in liver of Vps15-LKO mice.

Next, to address whether transcriptional activation of PPARα by fenofibrate was sufficient to rescue the lipid oxidation in livers of Vps15-LKO mice, we analysed relative metabolite levels by mass spectrometry. As expected, fenofibrate administration resulted in a increase of short-chain and sharp decrease of long-chain fatty acid carnitine derivatives in the livers of control mice (Fig. 3d). Notably, this two-week fenofibrate treatment increased Acetyl-CoA and normalized levels of long-chain fatty acid carnitine derivatives in livers of Vps15-LKO mice (Fig. 3d). Thus, pharmacologic activation of PPARα by a synthetic ligand is sufficient to rescue defective lipid oxidation in livers of Vps15-LKO mice.

**PPARα repressors are accumulated in livers of Vps15-LKO mice.** The defective PPARα transcriptional responses in livers of Vps15-LKO mice concurred with the nuclear accumulation of its bona fide repressors, Hdac3 and NCoR1 (Fig. 3c). It was also evident at total protein level in extracts of liver tissue of Vps15-LKO mice and increased complex formation of PPARα with NCoR1 protein detected in liver extracts of Vps15 mutants (Supplementary Fig. 5c and 5d). Notably, there were no difference in *NCoR1* and *Hdac3* transcript levels between the control and Vps15 mutant (Supplementary Fig. 5e). Moreover, fenofibrate treatment was sufficient to reduce nuclear levels of Hdac3 and NCoR1 repressors without affecting their total protein levels in livers of Vps15-LKO mice (Fig. 3c; Supplementary Fig. 5c).

Inhibition of the class 3 PI3K by targeting Vps15 in different cell types, including hepatocytes, results in lysosomal dysfunction[31,33]. In line, both in vivo and ex vivo autophagic flux analyses demonstrated that the protein levels of NCoR1 and Hdac3 proteins were increased with inhibition of lysosomal activity using chloroquine (raises the lysosomal pH) or leupeptin (lysosomal protease inhibitor) in livers of fasted wild-type mice unlike in Vps15 mutant (Fig. 3e; Supplementary Fig. 5f). The effective lysosomal flux under these conditions was evidenced by p62 degradation in livers of fasted mice and recovery of its levels

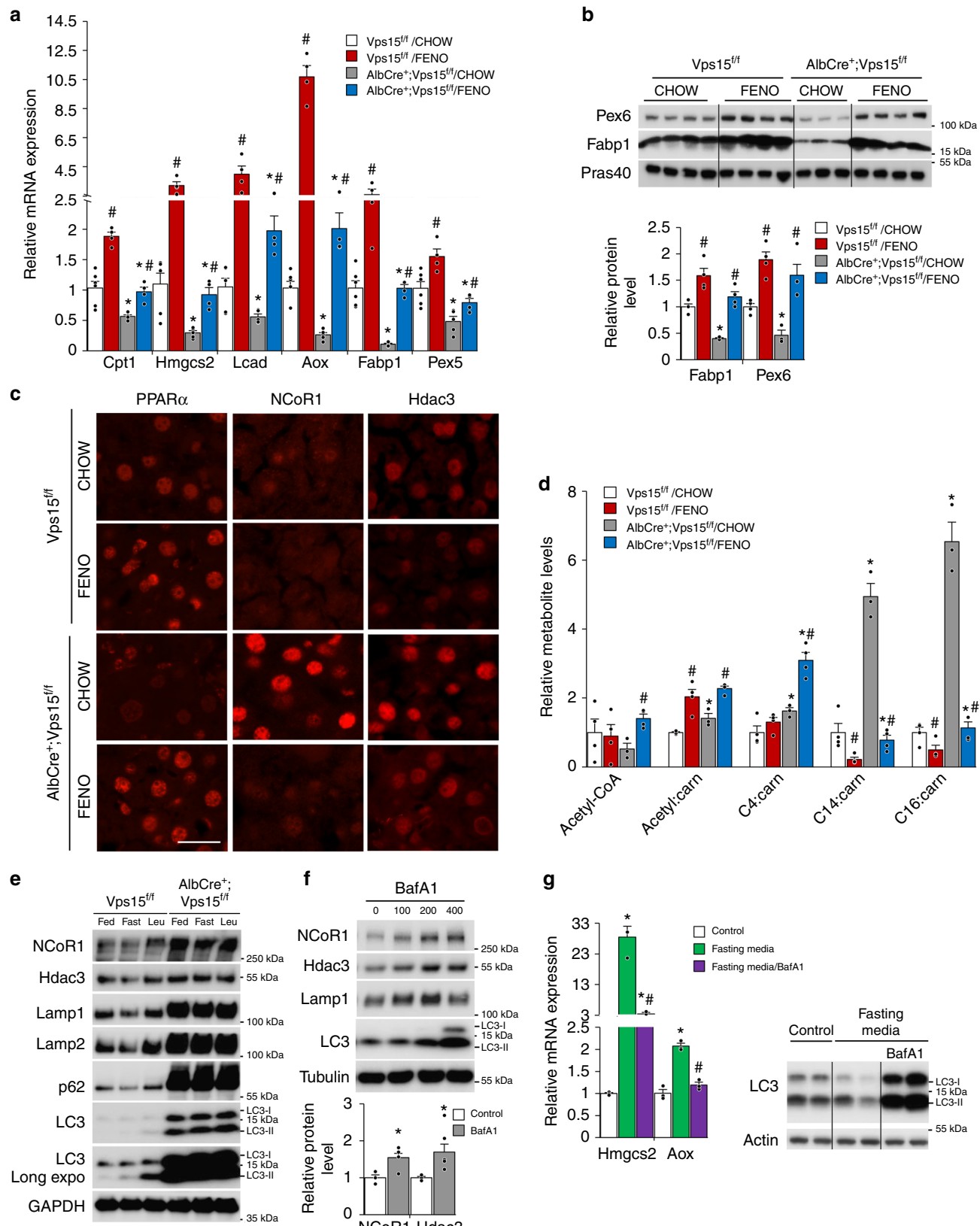

upon treatment with lysosomal inhibitors (Fig. 3e; Supplementary Fig. 5f). In addition, starvation either in vivo or ex vivo promoted lipidation of LC3 protein, signifying autophagy induction, a response that was further augmented when lysosomal activity was inhibited (Fig. 3e; Supplementary Fig. 5f). In line with blocked

autophagic flux in livers of Vps15-LKO mice, LC3 and p62 proteins were highly accumulated and their levels were not further modified by those treatments (Fig. 3e; Supplementary Fig. 5f). These observations in liver tissue were further confirmed by findings in primary hepatocytes. Likewise, the inhibition of

**Fig. 3** PPARα activation by fenofibrate restores lipid catabolism in Vps15-LKO mice. **a** Relative transcript levels of metabolic enzymes in ketogenesis, FAO, fatty acid transport and peroxisome biogenesis in livers of fed Vps15$^{f/f}$ and AlbCre$^+$;Vps15$^{f/f}$ mice that were treated for two weeks with fenofibrate. Data are means ± SEM (Vps15$^{f/f}$ ($n = 6$ and $n = 4$ for chow and FENO group), AlbCre$^+$;Vps15$^{f/f}$ ($n = 5$ and $n = 4$ for chow and FENO group), $P < 0.05$ *: vs Vps15$^{f/f}$, #: vs chow, two-tailed, unpaired Student's $t$ test). **b** Immunoblot analysis of total protein liver extracts of mice treated as in **a** using indicated antibodies. Densitometric analyses of protein levels normalised to Pras40 levels presented as folds over Vps15$^{f/f}$-chow condition. Data are means ± SEM ($n = 4$ for Vps15$^{f/f}$ chow and FENO group, $n = 3$ for AlbCre$^+$;Vps15$^{f/f}$ chow and $n = 4$ for AlbCre$^+$;Vps15$^{f/f}$ FENO group, $P < 0.05$ *: vs Vps15$^{f/f}$, #: vs chow, two-tailed, unpaired Student's $t$ test). **c** Representative images of immunofluorescent analyses showing nuclear localization of PPARα, NCoR1 and Hdac3 in liver tissue of Vps15$^{f/f}$ and AlbCre$^+$;Vps15$^{f/f}$ mice treated as in **a**. Secondary anti-mouse or anti-rabbit IgG Alexa Fluor 568 antibody were used for detection. Scale bar: 40 μm. **d** Relative levels of Acetyl-CoA and Acyl-carnitine metabolites measured by mass spectrometry in liver tissue of mice treated as in **a**. Data are means ± SEM (Vps15$^{f/f}$ ($n = 4$ for chow and FENO group), AlbCre$^+$;Vps15$^{f/f}$ ($n = 3$ and $n = 4$ for chow and FENO group), $P < 0.05$ *: vs Vps15$^{f/f}$, #: vs chow, two-tailed, unpaired Student's $t$ test). **e** Immunoblot analyses in total protein liver extracts of 6-week old Vps15$^{f/f}$ and AlbCre$^+$; Vps15$^{f/f}$ mice. Mice were either fed or fasted for 24 h. Four hours prior the sacrifice fasted mice were injected with leupeptin (40 mg/kg) or vehicle. The total protein extracts were immunoblotted with indicated antibodies. **f** Immunoblot analysis of cytosolic protein fractions of primary hepatocytes that were grown for 72 h in fasting media and treated for 24 h before collection with increasing doses of BafA1. Densitometric analyses of protein levels normalised to Tubulin levels presented as folds over vehicle-treated cells. Data are means ± SEM ($n = 4$ with 100 nM Bafilomycin A1 treatment, $P < 0.05$ *: vs vehicle, two-tailed, unpaired Student's $t$ test). **g** Relative transcript levels (left panel) of indicated genes in primary hepatocytes incubated in control or fasting media (72 h) treated with or without BafA1 for 24 h before collection. Data are means ± SEM ($n = 3$, $P < 0.05$ *: vs control media, #: vs fasting media, two-tailed, unpaired Student's $t$ test). The immunoblot (right panel) served as control of autophagic activity

lysosomal activity in primary hepatocytes with Bafilomycin A1, a selective inhibitor of the vacuolar proton pump, was sufficient to induce accumulation of Hdac3 and NCoR1 proteins, mimicking the observations in livers of Vps15-LKO mice (Fig. 3f). In addition, immunofluorescent analyses in livers of fasted wild-type mice showed decreased nuclear levels of Hdac3 and NCoR1 repressors (Supplementary Fig. 5g). Those were further corroborated by immunoblot analyses in liver tissue of 24-h fasted wild-type and Vps15-LKO mice. In line with transcriptional activation of PPARα by fasting, its nuclear protein levels were upregulated in livers of wild-type mice, a response largely lacking in Vps15 mutant (Supplementary Fig. 5h). This response in PPARα protein expression was paralleled by its transcript levels, consistent with auto-control of its expression (Supplementary Fig. 5i). These changes in PPARα protein were inversely correlated with nuclear Hdac3 and NCoR1 expression (Supplementary Fig. 5h). Consistent with lysosomal dysfunction in Vps15-null liver, the accumulation of NCoR1 and Hdac3 proteins was not rescued by fasting (Supplementary Fig. 5j). The inhibitory effect of lysosomal dysfunction on PPARα transcriptional activity was further evidenced by analyses in primary hepatocytes maintained in a fasting-mimicking media with or without Bafilomycin A1. Fasting-mimicking conditions led to autophagy induction, as judged by LC3 lipidation, and resulted in a potent induction of bona fide PPARα target genes in FAO (*Aox*) and ketogenesis (*Hmgcs2*) (Fig. 3g). Importantly, these transcriptional responses were abrogated by lysosomal inhibition with Bafilomycin A1 (Fig. 3g). Altogether, these observations back the hypothesis that an excess of PPARα repressors, due to their defective lysosomal degradation, and their association with PPARα contribute to PPARα inhibition in Vps15-LKO mice.

**PPARα transcriptional repressors are autophagy substrates.** The lysosomal degradation of PPARα transcriptional repressors was further corroborated by the immunofluorescent analyses showing targeting of endogenous NCoR1 and Hdac3 to lysosomal membrane protein Lamp2-positive cellular compartment (Fig. 4a). These were further supported by findings of NCoR1 protein co-localization with the lysosomal membrane protein Lamp1-positive structures under conditions of autophagic flux (Supplementary Fig. 6a). Autophagosome membrane localized proteins of Atg8-like family, gamma-aminobutyric acid receptor-associated protein (GABARAP) and LC3, act as receptors for selective autophagic degradation of autophagy substrates. Therefore, we asked whether Hdac3 and NCoR1 proteins could

bind to GABARAP and LC3. The GFP-tagged GABARAP and LC3 proteins were transiently expressed in HEK293T cells, a cell line in which endogenous Hdac3 and NCoR1 proteins are expressed and which is characterized by potent autophagic responses. The pull down of GFP-tagged LC3 and GABAPAP proteins using Trap-GFP agarose retrieved endogenous p62 protein, a known interacting partner of both proteins[46] (Fig. 4b). The selectivity of the assay was revealed by exclusive interaction of ULK1 with GABARAP but not with LC3 protein[47] (Fig. 4b). Notably, consistent with the requirement of the lipid kinase activity of the class 3 PI3K for autophagy, the binding of ULK1 and p62 with Atg8-like proteins was decreased in cells treated with the selective class 3 PI3K inhibitor, PIK-III (Fig. 4b). Importantly, both NCoR1 and Hdac3 were found in GABARAP and LC3 precipitates with higher affinity to GABARAP protein compared to LC3 protein (Fig. 4b). Notably, the binding of NCoR1 and Hdac3 to GABARAP was decreased by pharmaco-logical inhibition of the class 3 PI3K lipid kinase with PIK-III (Fig. 4b). Inversely, Vps15 overexpression promoted the binding of NCoR1 and Hdac3 to GABARAP (Fig. 4c). In the same conditions, Vps15 expression in HEK293T cells increased PPARα levels (Supplementary Fig. 6b). Next, we asked whether Vps15 overexpression would affect the interaction of PPARα and Hdac3. The co-immunoprecipitation analyses demonstrated that binding of Hdac3 to PPARα was reduced by ectopic expression of Vps15 (Fig. 4d). These biochemical analyses were further supported by the results of a proximity ligation assay which demonstrated decreased interaction of Hdac3 and PPARα in Vps15-expressing cells (Fig. 4e). In sum, these findings suggest that NCoR1 and Hdac3 co-repressors of PPARα degrade by autophagy through interaction with Atg8-like proteins in class 3 PI3K-dependent manner.

**Hdac inhibition rescues PPARα responses in Vps15-LKO mice.** Hdac3 is the enzymatically active component of the NCoR1-containing repressor complex. To further test the contribution of Hdac3 in PPARα repression and the mitochondrial dysfunction in Vps15-depleted hepatocytes, the Hdac3 levels were down-regulated by expressing selective shRNA in primary hepatocytes. Acute depletion of either Vps15 or Vp34 in primary hepatocytes resulted in inhibition of mitochondrial respiration, a defect that was rescued by the co-depletion of Hdac3 (Fig. 5a and Supplementary Fig. 7a). Similarly, the knockdown of Hdac3 in Vps15-null primary hepatocytes isolated from Vps15-LKO mice restored expression of PPARα targets, *Aox1* and *Cpt1b* (Supplementary

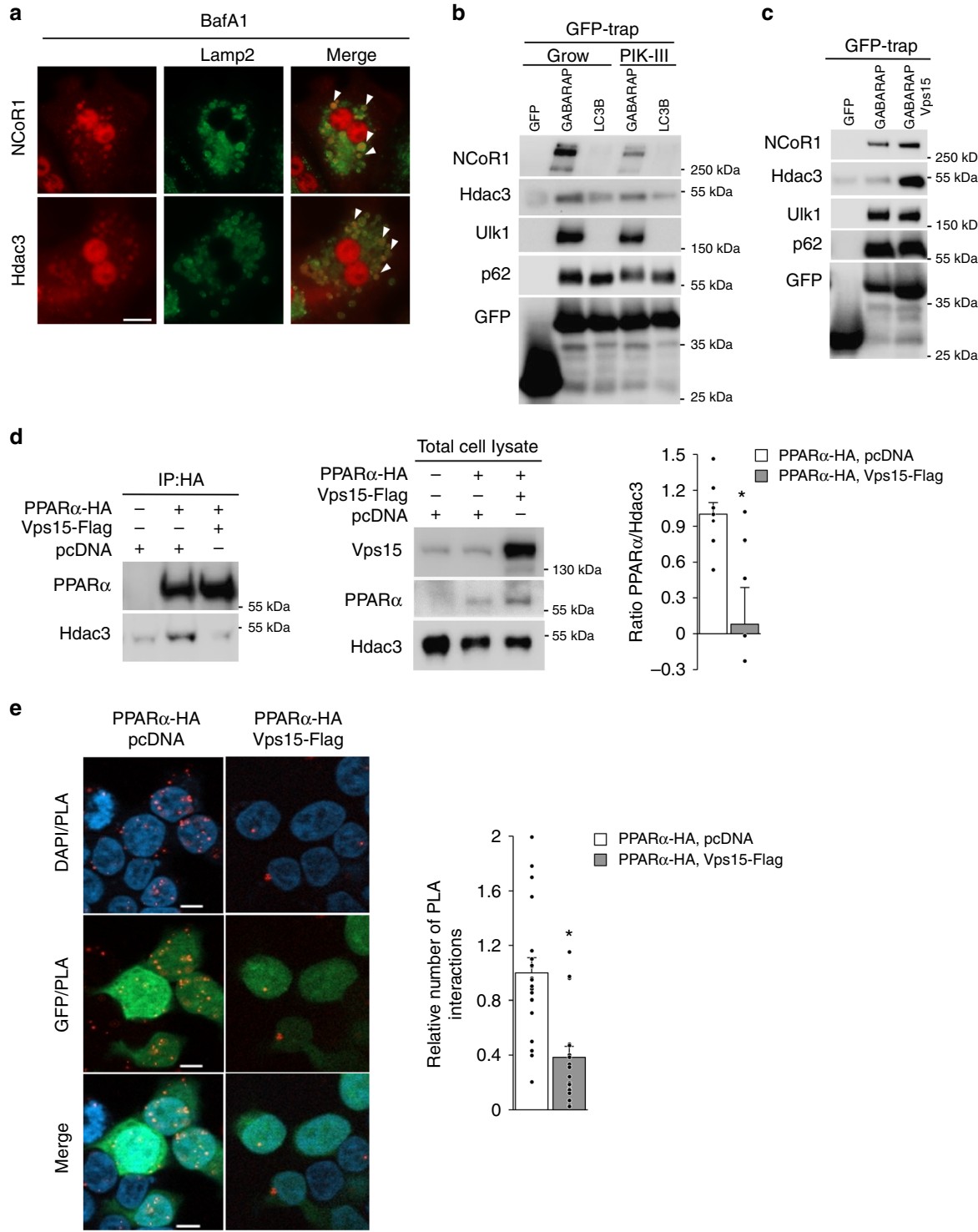

Fig. 7b and 7c). We extended these observations to in vivo by pharmacologically inhibiting histone deacetylases by clinically used drug valproic acid (VPA)[48]. Notably, we found that in addition to Hdac3, Hdac1 was also upregulated in livers of Vps15-LKO mice, further reasoning the use of the wide-range inhibitor of deacetylases, such as VPA (Supplementary Fig. 7d). The efficacy of VPA was evidenced by elevated Histone H3 acetylation on K27 in chromatin extracts of Vps15-LKO mice (Fig. 5b). Consistently, Hdac inhibition increased expression of PPARα targets in livers of the Vps15 mutant without modifying transcript or protein levels of PPARα and Hdac3 in livers of

Vps15-LKO mice (Figs. 5c, d and Supplementary Fig. 7d–7f). Importantly, VPA treatment resulted in a significant decrease of long-chain acyl-carnitine levels suggesting improved FAO in livers of Vps15-LKO mice (Fig. 5e). Altogether, these findings demonstrate that the repressive activity of Hdac3, at least partially contributes to PPARα inhibition in Vps15-LKO mice.

**PPARα activation restores mitochondria mass in Vps15 mutants.** Defective lipid oxidation and decreased mitochondria mass in hepatic *Vps15* mutant concurred with the down-regulation of the transcriptional programmes for mitochondria

**Fig. 4** Hdac3 and NCoR1 repressors degrade by autophagy. **a** NCoR1 and Hdac3 co-localize with Lamp2 protein in primary hepatocytes. Primary hepatocytes treated for 24 h with 100 nM BafA1 were PFA fixed and co-stained with primary antibodies, secondary anti-rabbit IgG Alexa Fluor 568 and anti-rat IgG Alexa Fluor 488 antibodies were used for detection. The white arrowheads indicate double-positive cytosolic structures. Scale bar: 40 μm. **b** Immunoblot analyses of GFP-Trap agarose eluates with indicated antibodies. HEK293T cells were transfected with GFP-GABARAP, GFP-LC3 or GFP expressing plasmid constructs. Twenty-four hours post-transfection cells were treated with Vps34 inhibitor PIK-III (2.5 μM) for 6 h before collection and proceeding with immunoprecipitation using GFP-Trap agarose. **c** Immunoblot analyses of GFP-Trap agarose eluates with indicated antibodies. HEK293T cells were transfected with GFP and GFP-GABARAP with or without Vps15-Flag expressing plasmid constructs. Twenty-four hours post-transfection cells were collected for immunoprecipitation using GFP-Trap agarose. **d** Immunoblot analyses of PPARα-containing complexes immunoprecipitated from HEK293T cells. HEK293T cells were transiently transfected with indicated plasmid constructs. Cells were collected 24 h post-transfection and PPARα complexes were immunoprecipitated using anti-HA antibody. The quantification of endogenous Hdac3 protein in PPARα immunoprecipitates normalized to the non-specific binding to beads is presented as fold change difference over empty vector-transfected condition. Data are means ± SEM ($n = 8$, $P < 0.05$ *: vs empty vector-transfected cells). **e** Proximity ligation assay between endogenous Hdac3 and ectopic PPARα-HA protein in HEK293T cells co-transfected with empty vector or Vps15-Flag expressing plasmid constructs (in all conditions GFP-expressing plasmid was co-transfected). PLA-detected PPARα/Hdac3 interactions (red dots) were quantified in $n = 100–150$ GFP-positive cells captured on $n = 19–21$ individual fields. The graph represents the number of PLA interactions as folds over empty vector-transfected condition. Data are means ± SEM ($P < 0.05$ *: vs empty vector-transfected cells). Scale bar: 10 μm

biogenesis and maintenance (Fig. 1). The ultrastructural analyses demonstrated that while the number of the individual mitochondria was not changed in Vps15-null hepatocytes, the mitochondrial area was decreased in hepatocytes of Vps15-LKO mice (Fig. 6a). Notably, fenofibrate treatment was sufficient to restore mitochondrial area in *Vps15*-null hepatocytes (Fig. 6a). In line with the findings of fragmented mitochondria in Vps15-null hepatocytes, the immunoblot analyses in total protein extracts of liver tissue suggested activated mitochondrial fission. Specifically, the phosphorylation of small GTPase dynamin-related protein 1 (DRP1) at Ser616, a read-out of induced mitochondrial fission, was upregulated in livers of Vps15-LKO mice (Supplementary Fig. 8a). This response was induced by a 24-h fasting in livers of wild-type mice and was constitutively upregulated in livers of Vps15-LKO mice (Supplementary Fig. 8a). It was also paralleled by stabilization and ubiquitination of Parkin E3 ubiquitin ligase that is important for selective autophagy of mitochondria (Supplementary Fig. 3d and 8a)[49]. Further immunoblot analyses have demonstrated that, consistent with induced by fasting mitochondrial fission and autophagy, Parkin and LC3-II were enriched in mitochondria fraction purified from livers of wild-type mice (Supplementary Fig. 8b). In line with mitochondrial damage, Parkin, LC3-II and poly-ubiquitinated protein levels were upregulated in purified mitochondria from livers of Vps15-LKO mice and were unmodified further by fasting (Supplementary Fig. 8b). Altogether, these ultrastructural and expression analyses suggest that mitochondrial dysfunction in Vps15-LKO mice is accompanied by induced mitochondrial fission resulting in fragmented mitochondrial network. The restoration of mitochondrial mass by fenofibrate in Vps15-null hepatocytes was further corroborated by findings of increased protein expression of the mitochondrial transcription factor (Tfb2m) and respiratory chain subunits (ND2, CYTB, ATP6) (Fig. 6b). These expression analyses were extended to show increased transcription of the nuclear coded mitochondria fusion protein (*Mfn2*), mitochondria DNA replication and transcription factors (*Tfam, Tfb2m, Polg*), as well as their immediate downstream targets, such as respiratory chain subunits (*ND1, ND2, Cytb, MTCO2, Atp6*) in livers of Vps15-LKO and control mice treated by fenofibrate (Fig. 6c). Altogether, these findings suggest that PPARα, in addition to its role in transcriptional induction of FAO, might have a function in mitochondrial biogenesis and maintenance.

**PGC1α rescues mitochondria activity in *Vps15*-null hepatocytes.** The transcription factor co-activator PGC1α plays a major role in mitochondrial biogenesis and metabolic activity by promoting the expression of nuclear encoded transcription factors

and enzymes that function in mitochondria[22]. It also co-operates with PPARα in the activation of genes involved in FAO[50,51]. PGC1α expression was downregulated both at the transcript and protein level in Vps15-LKO livers (Supplementary Fig. 8c and 8d). In agreement with its co-activating role for PPARα, ectopic expression of PGC1α potently activated transcriptional activity on a conventional luciferase reporter construct that contains three tandem-repeated PPAR response elements (3xPPRE-LUC) (Fig. 6d). It also resulted in a transcriptional upregulation of a PPARα target in FAO (Cpt1) as well as mitochondria biogenesis factors Tfam and Tfb2m (Fig. 6e). Importantly, the co-expression of Vps15 with PGC1α led to potentiation of these transcriptional responses (Fig. 6d, e). It was also paralleled by amplified autophagic degradation evidenced by decreased levels of bona fide autophagic substrate p62 protein and lower levels of LC3-II form as well as decreased Hdac3 levels (Supplementary Fig. 8e). This decreased Hdac3 expression was due to its lysosomal degradation as it was suppressed by inhibition of lysosomal activity with Bafilomycin A1 and Leupeptin (Supplementary Fig. 8f). Given a prominent role of PGC1α in mitochondrial biogenesis, we asked whether restoration of PGC1α expression would be sufficient to rescue the mitochondrial defects in *Vps15*-null hepatocytes. The acute depletion of Vps15 in primary hepatocytes resulted in decreased transcript levels of key mitochondria biogenesis factors (Tfam, Tfb1m and Polg) and known PPARα target Cpt1 (Fig. 6f). Importantly, adenoviral expression of PGC1α was sufficient to induce transcription of Tfam, Tfb1m, Polg and Cpt1 in primary hepatocytes of wild-type mice and induced their levels in *Vps15*-depleted hepatocytes (Fig. 6f). Furthermore, similar to the effect of Vps15 re-expression, PGC1α expression rescued the mitochondria respiration defect in *Vps15*-depleted hepatocytes (Fig. 6g). In sum, PGC1α is sufficient to restore mitochondrial biogenesis and activity in *Vps15*-null hepatocytes.

**PPARα activates mitochondrial biogenesis in liver.** Direct transcriptional regulation by PPARα is the most likely explanation for fenofibrate's effect on mitochondrial biogenesis. In line, activation of PPARα by fenofibrate or in fasting concurred with an upregulation of mitochondrial biogenesis factors and respiratory chain subunits in the livers of control mice in PPARα-dependent manner (Fig. 7a and Supplementary Data 5). These observations in microarray sets were further confirmed by RT-qPCR (Fig. 7b). These were in line with increased mitochondrial DNA content in livers of wild-type mice induced by fasting, a response abrogated in Vps15-LKO and PPARα-LKO mice (Fig. 7c, d). The implication of PPARα in mitochondrial biogenesis was further corroborated by findings in primary

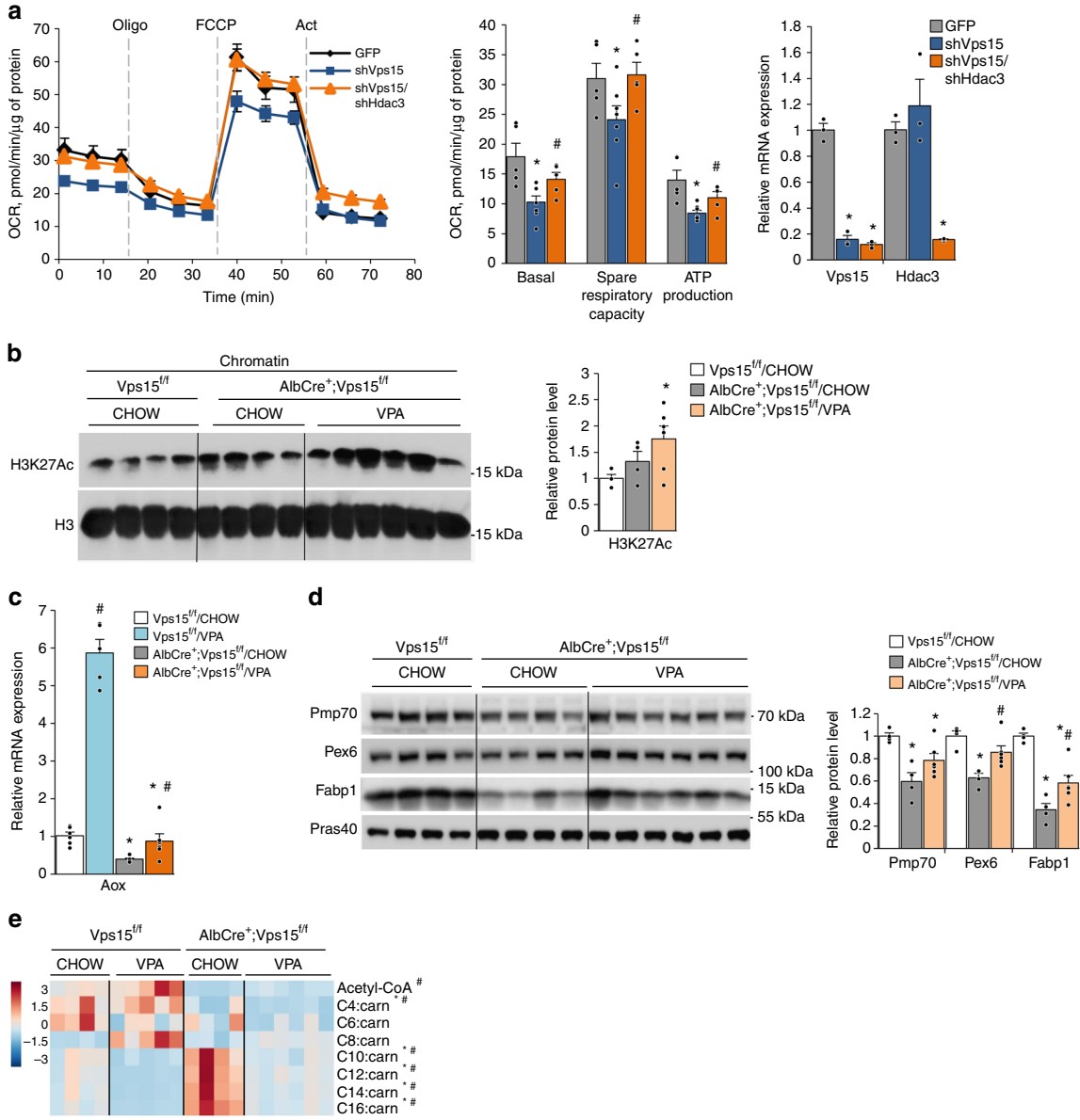

**Fig. 5** Inhibition of Hdac3 improves PPARα responses in livers of Vps15-LKO mice. **a** The OCR measured by SeaHorse Bioanalyzer in primary hepatocytes 48 h post-transduction with adenoviral vectors expressing GFP, shRNAVps15 or shRNAVps15 + shRNAHdac3 under basal conditions and in response to sequential treatment with indicated reagents. Dashed lines indicate the time of the addition of each reagent. Representative experiment of three independent hepatocyte cultures is presented. Quantification of basal respiration, ATP production and maximal respiratory capacity are shown on the graphs (middle panel). Relative transcript levels of *Vps15* and *Hdac3* is shown on right panel. Data are means ± SEM (*n* = 3, *P* < 0.05 *: vs GFP-infected cells, #: vs shRNAVps15-infected cells, two-tailed, unpaired Student's *t* test). **b** Immunoblot analysis of chromatin extracts from the liver tissue of six week old random-fed Vps15^f/f and AlbCre^+;Vps15^f/f mice that were treated for two weeks with VPA incorporated in food or control chow food. Densitometric analyses of H3K27Ac levels normalised to total H3 levels presented as folds over Vps15^f/f-chow condition. Data are means ± SEM (*n* = 4 for Vps15^f/f and AlbCre^+;Vps15^f/f chow, *n* = 6 AlbCre^+;Vps15^f/f VPA, *P* < 0.05 *: vs Vps15^f/f, #: vs chow, two-tailed, unpaired Student's *t* test. **c** Relative mRNA expression levels of *Aox* in the livers of mice treated as in **b**. Data are means ± SEM (*n* = 6 for Vps15^f/f and AlbCre^+;Vps15^f/f chow, *n* = 5 Vps15^f/f VPA, *n* = 6 AlbCre^+; Vps15^f/f VPA, *P* < 0.05 *: vs Vps15^f/f, #: vs chow, two-tailed, unpaired Student's *t* test). **d** Immunoblot analysis of total protein liver extracts of mice treated as in **b** using indicated antibodies. Densitometric analyses of protein levels normalised to Pras40 levels presented as folds over Vps15^f/f-chow condition. Data are means ± SEM (*n* = 4 for Vps15^f/f and AlbCre^+;Vps15^f/f chow, *n* = 6 AlbCre^+;Vps15^f/f VPA, *P* < 0.05 *: vs Vps15^f/f, #: vs chow, two-tailed, unpaired Student's *t* test). **e** Heat map showing relative levels of Acetyl-CoA and Acyl-carnitine metabolites measured by mass spectrometry in the livers of mice treated as in **b**. Each column corresponds to an individual animal and the colour of the cell indicates the relative content of the metabolite (from blue to red). Data are means ± SEM (*n* = 4 for Vps15^f/f and AlbCre^+;Vps15^f/f chow, *n* = 5 Vps15^f/f VPA, *n* = 6 AlbCre^+;Vps15^f/f VPA, *P* < 0.05 *: AlbCre^+; Vps15^f/f vs Vps15^f/f, #: AlbCre^+;Vps15^f/f VPA vs AlbCre^+;Vps15^f/f chow, two-tailed, unpaired Student's *t* test)

hepatocytes. As expected, incubation in fasting media stimulated expression of bona fide PPARα target genes involved in FAO (*Aox*), ketogenesis (*Hmgcs2*) and peroxisome function (*Pex6*) (Fig. 7e, f). Importantly, it was sufficient to promote expression of

mitochondrial transcription factors and their downstream targets (Fig. 7e, f). These findings suggested that PPARα might directly activate transcription of mitochondrial biogenesis factors. To test this hypothesis, we performed a bioinformatics analysis of *Tfam*

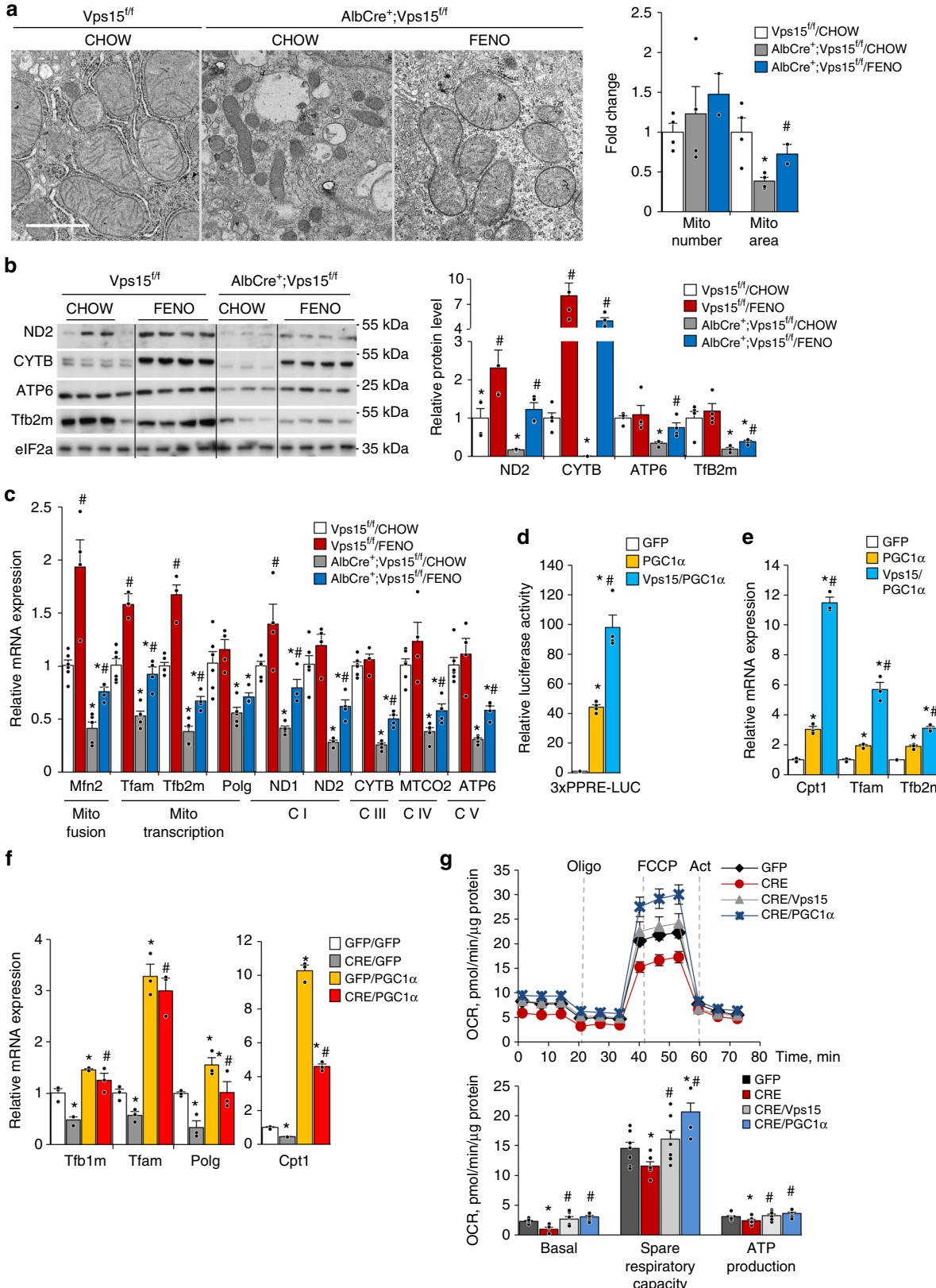

and *Tfb2m* gene promoters for the presence of putative PPREs. These in silico analyses identified putative PPREs that were localized in the gene of *Tfam* (PPRE#1 at −277 bp and PPRE#2 at +660 bp from the transcription start site) and in the gene of *Tfb2m* (PPRE#1 at −361 bp, PPRE#2 at +181 bp from the transcription start site) (Fig. 7g). To evaluate if PPARα directly

binds to those putative PPREs, we performed chromatin immunoprecipitation (ChIP) with endogenously expressed PPARα and analyzed enrichment in PPARα binding using primers nested around the identified PPRE sequences. As shown in Fig. 7g, in mouse primary hepatocytes, endogenous PPARα was bound to promoters of both genes and its binding was similar to a

**Fig. 6** Fenofibrate treatment rescues mitochondrial function in *Vps15*-LKO mice. **a** Representative liver transmission electron micrographs of control or fenofibrate treated Vps15$^{f/f}$ and AlbCre$^+$;Vps15$^{f/f}$ mice. The fold change of relative mitochondria area and mitochondria number are presented as graph. Data are means ± SEM of calculations of total ~1500 mitochondria normalized to total area of the fields analysed (n = 15–20 randomly taken fields, n = 2–4 mice, P < 0.05 *: vs Vps15$^{f/f}$, #: vs chow, two-tailed, unpaired Student's t test). Scale bar: 1 μm. Protein (**b**) and relative transcript levels (**c**) of mitochondria fusion factor, mitochondrial DNA transcription factors, mitochondrial DNA polymerase and respiratory chain subunits in liver of Vps15$^{f/f}$ and AlbCre$^+$;Vps15$^{f/f}$ mice treated with fenofibrate. Densitometric analyses of protein levels normalised to eIF2a levels presented as folds over Vps15$^{f/f}$-chow condition. Data are means ± SEM (n = 4–6 for Vps15$^{f/f}$ chow and FENO group, n = 4–5 for AlbCre$^+$;Vps15$^{f/f}$ chow and n = 4 for AlbCre$^+$;Vps15$^{f/f}$ FENO group, P < 0.05 *: vs Vps15$^{f/f}$, #: vs chow, two-tailed, unpaired Student's t test). **d** Primary hepatocytes were transduced with adenoviral vectors expressing PGC1α, PGC1α + Vps15 or GFP protein as a control and 24 h post-transduction were transfected with reporter construct. The relative luminescence presented as fold difference over empty vector transfected GFP-infected condition. Data are means ± SEM (n = 4 independent hepatocyte cultures, P < 0.05 *: vs empty vector, #: vs PGC1α, two-tailed, unpaired Student's t test). **e** Relative mRNA expression levels of indicated genes in primary hepatocytes 36 h post-transduction with indicated adenoviral vectors. Data are means ± SEM (representative experiment of n = 3 independent cultures, P < 0.05 *: vs GFP, #: vs PGC1α, 2-tailed, unpaired Student's t test). **f** Relative transcript levels of mitochondria biogenesis factors and PPARα target *Cpt1* in primary Vps15$^{f/f}$ hepatocytes in which Vps15 was depleted by expressing Cre recombinase using adenoviral vectors. Thirty-six hours after initial infection PGC1α was expressed for additional 24 h using adenoviral vectors. Data are means ± SEM (representative experiment of n = 3 independent hepatocyte cultures, P < 0.05 *: vs GFP/GFP, #: vs Cre/GFP, two-tailed, unpaired Student's t test). **g** The OCR measured by SeaHorse Bioanalyzer in primary Vps15$^{f/f}$ hepatocytes in which Vps15 was depleted by expressing Cre recombinase using adenoviral vectors. Thirty-six hours after initial infection, cells were transduced with adenoviral vectors to express PGC1α or Vps15 for additional 24 h. First, OCR under basal conditions and then in response to sequential treatments is shown. Representative experiment of three independent hepatocyte cultures is presented. Dashed lines indicate the time of the addition of each reagent. Quantification of basal respiration, ATP production and maximal respiratory capacity are shown on the graphs (right panel). Data are means ± SEM (P < 0.05 *: vs GFP-infected cells, #: vs Cre-infected cells, two-tailed, unpaired Student's t test)

characterized PPRE in the promoter of its own gene. In addition, chromatin immunoprecipitation analyses from liver tissue demonstrated significantly lower enrichment of transcription permissive H3K27Ac histone mark in regions encompassing characterized PPREs in promoter of *PPARA*, its known target gene *Fabp1* as well as PPRE in the promoter of *Tfb2m* gene in livers of Vps15 mutants compared to controls (Supplementary Fig. 8g). These finding were consistent with decreased transcript levels of PPARα, Fabp1 and TFb2m and accumulation of PPARα transcriptional repressors, NCoR1 and Hdac3, in livers of Vps15-LKO mice. Altogether, these observations advocate that PPARα in addition to its established role in fatty acid degradation directly contributes to mitochondrial biogenesis in liver (Fig. 7h).

## Discussion

In growth permissive conditions with high nutrient levels, pro-anabolic insulin and mammalian target of rapamycin (mTOR) signalling inhibits autophagy and mitochondrial lipid catabolism[52]. Mechanistically, this involves the cytosolic retention and proteasomal degradation of TFEB and PPARα/PGC1α transcription complexes as well as the control of stability, nuclear localization and recruitment to transcription factors of NCoR1 and Hdac3 repressors[38,53–55]. The positive role of TFEB and PPARα in the transcriptional control of lysosomal activity and autophagy in liver was demonstrated during physiologic fasting when nutrient-driven insulin and mTOR signalling are blunted[17,20]. This synchronous activation of PPARα and PGC1α in fasting assures expression of genes involved in fatty acid transport, degradation, as well as autophagy machinery and mitochondrial biogenesis factors to match increased energy demand with the metabolic capacity of mitochondria. Notably, lysosomal pathway of autophagy is a central mechanism for the adaptation to fasting, as evidenced by perinatal lethality of autophagy deficient mice[3,4]. Yet, the possibility that autophagy controls at transcriptional level metabolic responses downstream of PPARα has not been addressed.

Among components of the autophagic pathway, the class 3 PI3K is ideally positioned to integrate signals from environment to adjust cellular metabolic activities depending on the energy status. Its lipid kinase activity drives autophagic flux and vesicular trafficking towards lysosomes, which are impaired in *Vps15* and *Vps34* tissue-specific mouse mutants[13,31,33,56,57]. In

addition to autophagy, the class 3 PI3K also assures a retro-control of insulin signalling by promoting lysosomal degradation of the insulin receptor in hepatocytes; and therefore, it impacts whole-body glucose homeostasis[33]. The findings that we present here in the context of liver tissue establish an additional feed-forward loop that couples activation of the class 3 PI3K to the transcriptional induction of autophagy, mitochondrial biogenesis and lipid catabolism downstream of PPARα. The most significant conclusions of our work are: (1) mitochondrial biogenesis and fatty acid oxidation in hepatocytes require functional class 3 PI3K; (2) liver-specific deletion of the regulatory subunit Vps15 phenocopies the loss of PPARα, suggesting that both genes functionally interact; (3) mechanistically, loss of the class 3 PI3K signaling in liver manifests in PPARα ligand shortage, decreased expression of its co-activator PGC1α and accumulation due to defective autophagic degradation of its transcriptional co-repressors Hdac3 and NCoR1. Altogether, our work demonstrates a key role of the class 3 PI3K in coordination of metabolic adaptation to fasting.

Likely, multiple mechanisms are involved in PPARα dysfunction in the class 3 PI3K mutant and future studies will establish their specific contribution. Our work suggests that, first, the defective lysosomal degradation of NCoR1 and Hdac3 repressors due to block of autophagy correlates with and probably contributes to PPARα inhibition in Vps15-LKO livers. Second, a lack of free fatty acids, which act as PPARα ligands, likely adds to PPARα dysfunction in the *Vps15* mutants. To this end, the inhibition of de novo fatty acid synthesis, a physiologically relevant source of PPARα ligands, would be consistent with the accumulation of Hdac3 and NCoR1 co-repressors which, in addition to PPARα, also inhibit pro-lipogenic liver X receptor (LXR) transcription factors[58]. Finally, the deletion of Vps15 in liver manifests in profound trafficking defects of plasma membrane receptors[33]. In fasting, adipose tissue derived fatty acids contribute as ligands to PPARα activation. Notably, the functional connection between mitochondria and endocytosis was suggested by a genome-wide RNAi screen of endocytosis regulators which revealed an enrichment in genes having mitochondrial-related functions[59]. Therefore, it is likely that in fasted Vps15-LKO mice, defective trafficking of the low-density lipoprotein receptor and lipid scavenger receptors, such as CD36, contribute to the lack of PPARα ligands delivered from periphery.

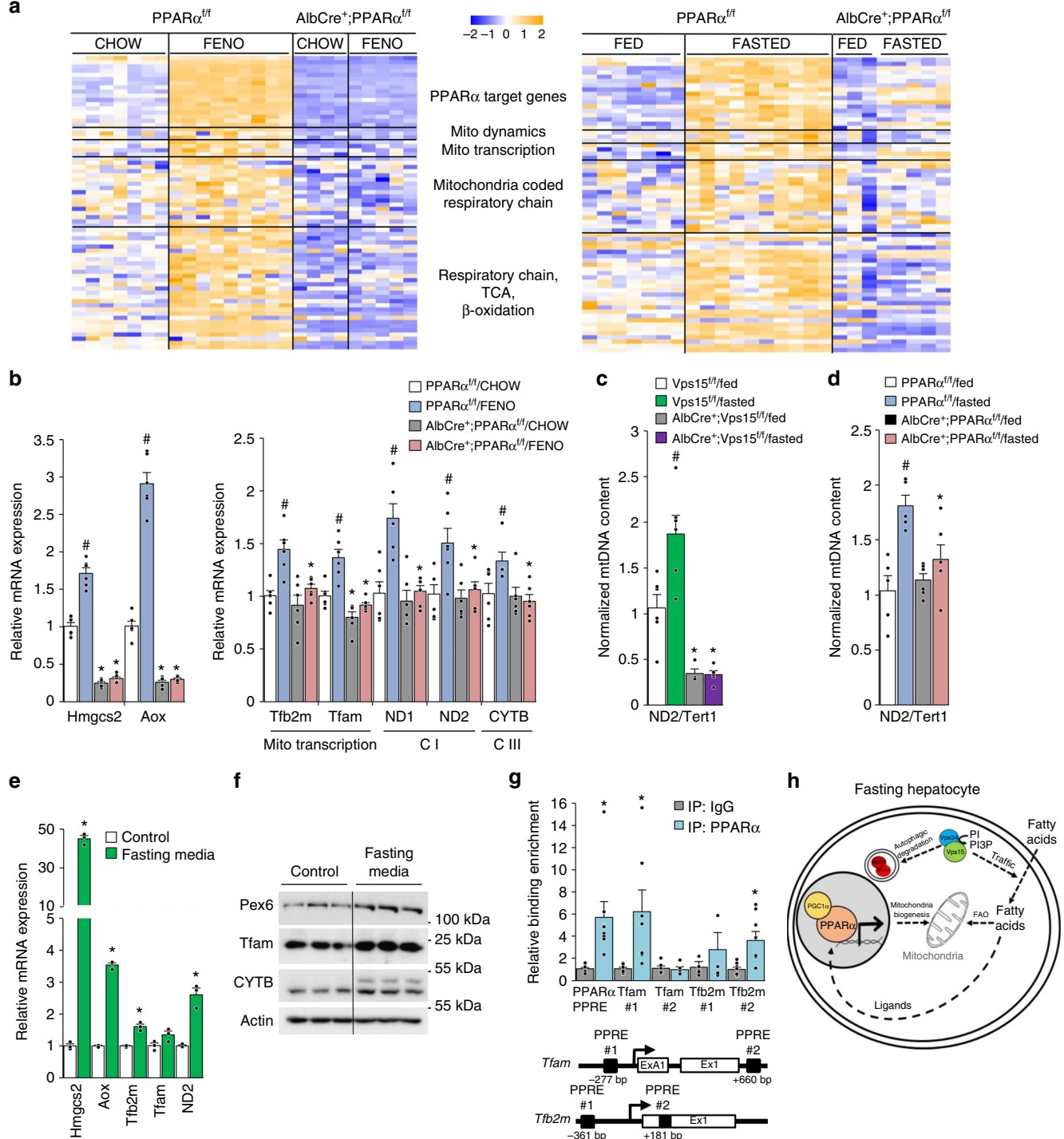

We find that hepatic PPARα, in addition to its known role in transcriptional activation of autophagy and lipid degradation, promotes mitochondrial biogenesis. This mechanism may ensure that an increased demand in the production of energy during fasting is coordinated with mitochondrial content. Notably, PGC1α levels, similar to PPARα, are decreased in livers of Vps15-LKO mice, and its expression is sufficient to restore mitochondrial dysfunction in *Vps15*-null hepatocytes. In addition, we cannot exclude that activated mitochondrial fission might contribute to mitochondrial dysfunction in livers of Vps15-LKO mice. Yet, likely, it is a secondary response to defective

mitochondrial turnover and mitochondrial biogenesis in hepatic Vps15 mutant. The positive function of PPARα in mitochondrial biogenesis might be restricted to hepatocytes. In the heart, PPARα activation inhibits respiratory chain gene expression, while in skeletal muscles, it promotes the metabolic transition towards glycolytic fibres which are characterised by a lower mitochondrial activity[60,61]. In skeletal muscles, deletion of PPARα reduces FAO but increases the number of oxidative fibres[61,62]. Mitochondrial biogenesis in the heart and skeletal muscles is induced by a transcriptional complex of ERR and PGC1α[60]. The molecular mechanisms of these tissue-specific

**Fig. 7** PPARα activates mitochondrial biogenesis in liver. **a** Heat map representing data from a microarray experiment performed with liver samples of PPARα[f/f] and AlbCre[+];PPARα[f/f] mice treated with fenofibrate (GSE73298) or fasted for 24 h (GSE73299). The colour of the cell indicates the relative change of expression (from blue to yellow). The genes are grouped according to their GO annotations (biological function). The gene names corresponding to GO groups are listed in Supplementary Data 5. **b** Relative mRNA expression levels of mitochondrial transcription factors, respiratory chain subunits and genes of FAO in liver tissue of PPARα[f/f] and AlbCre[+];PPARα[f/f] mice treated with fenofibrate[37]. Data are means ± SEM ($n = 6$, $P < 0.05$ *: vs PPARα[f/f], #: vs chow, two-tailed, unpaired Student's $t$ test). Mitochondrial versus nuclear DNA was determined by quantitative PCR analysed in total DNA purified from liver tissue of fed or 24-h fasted Vps15[f/f] and AlbCre[+];Vps15[f/f] mice (**c**) or PPARα[f/f] and AlbCre[+];PPARα[f/f] mice[37] (**d**). Mitochondrial genome coded ND2 gene served as read-out of mtDNA and Tert1 as read-out of nuclear genome. Data are means ± SEM ($n = 4-6$ for Vps15[f/f], $n = 4-5$ for AlbCre[+];Vps15[f/f], $n = 5-6$ for PPARα[f/f] and AlbCre[+];PPARα[f/f]; $P < 0.05$ *: vs wild-type mice, #: vs fed, two-tailed, unpaired Student's $t$ test). Relative mRNA (**e**) and protein (**f**) expression levels of indicated genes in primary hepatocytes incubated for 72 h in control or fasting media. Data are means ± SEM ($n = 3$, $P < 0.05$ *: vs control media, two-tailed, unpaired Student's $t$ test). **g** Schematic representation of putative PPRE localization in the promoter regions of mouse *Tfam* and *Tfb2m* genes (top panel). White rectangles represent exons, black rectangles—putative PPREs. Relative enrichment of endogenous PPARα on putative PPREs in promoter region of *Tfam* and *Tfb2m* genes analysed by qPCR on chromatin prepared from primary hepatocytes (bottom panel). Binding of PPARα to PPRE in its own promoter served as a positive control. Data are means ± SEM, average of $n = 7-8$ immunoprecipitations. $P < 0.05$ *: vs IgG, two-tailed, unpaired Student's $t$ test. **h** Proposed model depicting the class 3 PI3K which controls PPARα activation for lipid degradation and mitochondrial biogenesis during fasting

roles of PPARα in mitochondrial metabolism and the involvement of the class 3 PI3K await elucidation. Finally, given their global role in control of gene expression, accumulation of NCoR1 and Hdac3 repressors in livers of Vps15 mutants likely results in transcriptional alterations that go beyond inhibition of PPARα. Future studies will build a full molecular picture of transcriptional control downstream of class 3 PI3K in maintenance of metabolic homeostasis. In conclusion, we propose that during the feeding-fasting transition, the class 3 PI3K orchestrates liver energy metabolism through the control of endocytosis, lysosomal pathway of autophagy and transcriptional control of mitochondrial metabolism. This makes the class 3 PI3K an exquisite energy sensor that might exert transcriptional control of nuclear receptors in other organs beyond the liver. Thus, our findings could be relevant to an array of metabolic processes controlled by these ubiquitous nutrient sensors and might have a far-reaching impact in human pathophysiology.

## Methods

**Reagents**. The following primary antibodies were used: Vps15 (1:1000, Abnova, H00030849-M03; 1:1000, Genetex, GTX108953), p62 (1:1000, Abnova, H00008878-MO1), β-actin (1:5000, Sigma, A5316), Tubulin (1:1000, Sigma, T9026), HA (1:1000, 1:200 for IF, Sigma, H9658), Pmp70 (1:500, Sigma, SAB4200181), β-catenin (1:500, BD Biosciences, 610153), Cytochrome C (BD Biosciences, 556432), Pras40 (1:1000, Cell Signaling, 2691), Ulk1(1:1000, Cell signaling, 8054S), Lamin A/C (1:1000, Cell Signaling, 2032), NCoR1 (1:1000, Cell Signaling, 5948), Hdac3 (1:1000, Cell Signaling, 85057), H3K27Ac (1:1000, Cell Signaling, 8173), H3 (1:1000, Cell Signaling, 4499), Ulk1(1:1000, Cell signaling, 8054S), Ubiquitin (1:1000, Cell Signaling, 3936), pS616 Drp1 (1:1000, Cell Signaling, 3455), Drp1(1:1000, Cell Signaling, 8570), LC3xp (1:500, Cell Signaling, 3868), LC3 (1:1000, NanoTools, 0231–100/LC3–3–5–5F10), eIF2α (1:1000, Santa Cruz, sc-11386), PPARα (1:500, Santa Cruz, sc-398394, sc-9000), Tfb2m (1:500, Santa Cruz, sc-517095), GAPDH (1:1000, Santa Cruz, SC-25778), Nrf2 (1:1000, Santa Cruz, sc-13032), Parkin (1:1000, Santa Cruz, sc32282), GST (1:1000, Santa Cruz, sc-459), Pex6 (1:500, Santa Cruz, sc-271813), Fabp1 (1:500, Santa Cruz, sc-50380), NCoR1 (1:500, sc-515934, Santa Cruz), Cre (1:1000, GTX127270, Gene-Tex), Tom40 (1:1000, ProteinTech, 18409–1-AP), ND2 (1:500, Proteintech, 19704–1-AP), Cytb (1:500, Proteintech, 55090–1-AP), ATP6 (1:500, Proteintech, 55313–1-AP), PGC1α (1:1000, Proteintech, 66369–1-Ig), Hdac1 (1:1000, Thermo, PA1–860), Lamp1 (1:1000, Abcam, ab24170), Lamp2 (1:1000, Abcam, ab13524), GFP (1:1000, Clontech, 8362–1), Huwe1 (1:1000, Cell Signaling, 5695). Tube 2 agarose was from LifeSensors (UM402). GFP, GFP-Cre, Vps15 and shRNAVps15 expressing adenoviral vectors were described previously[31]. Adenovirus expressing shRNA Vps34 and Hdac3 were from Vector Biolabs. Adenovirus expressing PGC1α was kindly provided by Bertrand Blondeau. Plasmids expressing GFP-LC3B and GFP-GABARAP were kindly provided by Patrice Codogno. Plasmid expressing Vps15-Flag was purchased from MRC PPU Reagents and Services. Plasmid expressing PPARα-HA was kindly provided by Urs Albrecht.

**Animals**. The Vsp15 conditional mutant mouse line was established at the MCI/ICS (Mouse Clinical Institute - Institute Clinique de la Souris, Illkirch, France) as described[31]. Liver specific *Vps15* knockout mouse line was generated as described[33].

Mice were housed in specific pathogen-free conditions. Male mice (6–8 week old) were used for the experimentation. Mice were randomly allocated to experimental groups and at least three animals were used for each condition (as indicated in figure legends) to ensure statistics analyses. Animal numbers were chosen to reflect the expected magnitude of response taking into account the variability observed in previous experiments. All animal studies were performed by authorized users in compliance with ethical regulations for animal testing and research. The study was approved by the Direction Départementale des Services Vétérinaires, Préfecture de Police, Paris, France (authorization number 75-1313) and the ethical committee of Paris Descartes University (authorization number 17-052).

**Treatments and metabolic studies in vivo**. All animals used in the study were fed ad libitum standard chow diet (Teklad global protein diet; 20% protein, 75% carbohydrate, 5% fat) and kept under 12 h/12 h (8 am/8 pm) light on/off cycle. Animals were sacrificed between 2 and 4 pm unless indicated. For fasting experiment, the mice were food deprived for 24 h starting at 10 pm. For 6-h starvation for metabolomics analyses in liver tissue, the fasting was initiated at 8 am and liver tissue collected at 2 pm. VPA (200 mg/kg) and Fenofibrate (200 mg/kg) were incorporated in chow food and mice treated during two weeks with free access to control and drug incorporated food. Body composition was assessed in all mice using DEXA scan on the minispec LF50 Mq 7.5 NMR Analyzer (Brucker) according to manufacturer's instructions. Plasmatic levels of lactate, non-esterified fatty acids, triglycerides, glycerol and hydroxybutyrate were measured enzymatically using Olympus AU 400 apparatus. TG levels in the acetone extracts of liver tissue were determined using Triglycerides FS Kit (Diasys) according to the manufacturer's instructions and as described[63].

**Electron microscopy**. For ultrastructural analyses liver samples were prepared as described[64]. Briefly, liver was perfused first with PBS and then 4%PFA before dissection. Tissue was postfixed in 2% glutaraldehyde in 0.12 M phosphate buffer. After overnight fixation, tissues were treated with 1% osmium tetroxide in 0.12 M phosphate buffer and embedded in epon (Fluka). Ultrathin sections (70 nm) were cut and stained with uranyl acetate (Plano GmbH) and lead citrate (Electron Microscopy Sciences). Samples were analyzed with a transmission electron microscope (EM 902; ZEISS or CM10; Phillips) at an acceleration voltage of 80 kV, and pictures were acquired using a 2K-CCD (TRS Albert Tröndle) or SC200W (Orius) camera. Prior to quantification, a randomized blinded code was assigned by a separate researcher and the randomization was decoded at the time of the final data analysis.

**Cell culture**. Primary hepatocytes from 6 to 8 week-old mice were isolated by liver perfusion as described previously[33]. Hepatocytes were plated at $12 \times 10^4$ cells/cm$^2$ in Williams medium (Life Technologies) supplemented with 20% FBS, 100 nM insulin, 25 nM dexamethasone, penicillin (100 U/ml), streptomycin (100 μg/ml), and amphotericin B (Fungizone) (250 ng/ml). For the infections Vps15[f/f] hepatocytes 12 h after plating were infected with 10 MOI of adenoviral vectors, 2 h after addition of viral particles the media was changed to Williams medium supplemented with a mix of antibiotics and 25 nM dexamethasone. Cells were collected 24–48 h later for the analyses. For fasting-mimicking conditions, 12 h after plating, primary hepatocytes were washed with PBS and incubated for 72 h in Williams media devoid of serum and insulin, containing 50 μM WY-14643, 2 μM sodium octanoate, 25 nM dexamethasone, penicillin (100 U/ml), streptomycin (100 μg/ml) and amphotericin B (Fungizone) (250 ng/ml). For lysosomal inhibition, primary hepatocytes were treated for 24 h with 100 nM BafA1 before collection. For luciferase reporter assay, the reporter constructs were a kind gift of Dr. L. Fajas (pGL3

empty vector and pGL3–3xPPRE). Primary hepatocytes were transduced with adenoviral vectors and 24 h post transduction, cells were transfected with a mix of luciferase reporters and control plasmid expressing β-Galactosidase using Lipofectamine LTX (Invitrogen). Twenty-four hours post-transfection, cells were collected for luciferase reporter activity assay as described[65].

**Mitochondrial activity measurements.** For measurements of oxygen consumption rate by Seahorse bioanalyzer, primary hepatocytes were seeded at a density of $2 \times 10^4$ cells per well in a collagen coated XFe96 cell culture microplate. Twelve hours post-plating cells were infected with relevant adenoviruses and mitochondrial activity was assessed at the times indicated in figure legends. Before measurement cells were balanced for 1 h in unbuffered XF assay media (Agilent Technologies) supplemented for OCR analysis with 2 mM Glutamine, 10 mM Glucose and 1 mM Sodium Pyruvate. For OCR measurements, compounds were injected during the assay at the following final concentrations: Oligomycin (ATP synthase inhibitor to measure respiration associated with cellular ATP production, 1 μM), FCCP (uncoupling agent to measure the maximal respiration capacity; 1 μM), Rotenone and Antimycin A (ETC inhibitors to measure the non-mitochondrial respiration; 1 μM). The data were normalized to protein content measured in each well using BCA assay (Thermo Fisher Scientific) according to manufacturer's instructions. Respiratory chain enzyme activities and activity of lactate dehydrogenase were spectrophotometrically measured using a Cary 50 UV–visible spectrophotometer (Varian Inc, Les Ulis, France) as previously reported[66].

**Subcellular fractionation.** Nuclear and cytosolic fractions were prepared using NE-PER Kit (Pierce) according to manufacturer's recommendations from $1 \times 10^6$ cells or 50 mg of liver tissue. The chromatin fraction was extracted from insoluble nuclear pellet using acidic buffer as in ref. [65]. The crude mitochondria fraction from liver tissue was prepared as previously reported[67].

**Targeted metabolomics.** Targeted metabolomics analyses were performed as described[68]. Briefly, extraction solution used was 50% methanol, 30% ACN, and 20% water. The volume of extraction solution added was calculated from weight of powdered tissue (60 mg/ml). After addition of extraction solution, samples were vortexed for 5 min at 4 °C, and then centrifuged at $16,000 \times g$ for 15 min at 4 °C. The supernatants were collected and analyzed by liquid chromatography–mass spectrometry using SeQuant ZIC-pHilic column (Merck) for the liquid chromatography separation. Mobile phase A consisted of 20 mM ammonium carbonate plus 0.1% ammonia hydroxide in water. Mobile phase B consisted of ACN. The flow rate was kept at 100 ml/min, and the gradient was 0 min, 80% of B; 30 min, 20% of B; 31 min, 80% of B; and 45 min, 80% of B. The mass spectrometer (QExactive Orbitrap, Thermo Fisher Scientific) was operated in a polarity switching mode and metabolites were identified using TraceFinder Software (Thermo Fisher Scientific). For analyses, metabolomics data were normalized using the median normalization method. MetaboAnalyst 4.0 software was used to conduct statistical analyses and heatmaps generation, and unpaired two-sample $t$ test was chosen to perform the comparisons. The algorithm for heatmap clustering was based on the Pearson distance measure for similarity and the Ward linkage method for biotype clustering.

**Microscopy.** For fluorescent microscopy analyses on cells, primary hepatocytes were grown on collagen treated coverslips (Millipore). Twelve hours post-plating primary hepatocytes were fixed with 4% PFA in PBS for 20 min and permeabilized with 0.1% saponin in PBS for 10 min, followed by blocking in 3% BSA in PBS. Slides were treated with primary antibodies overnight. Secondary antibodies used for these assays were anti-rabbit IgG Alexa Fluor 568 or anti-rat IgG Alexa Fluor 488 (Life Technologies). For fluorescent microscopy analyses of PPARα in liver tissue, liver was fixed by perfusion with 4%PFA before dissection. The liver tissue samples were postfixed overnight in phosphate-buffered 10% formalin and embedded in paraffin. In total, 4-μm sections were cut and processed for staining with anti-PPARα antibody. For NCoR1 and Hdac3 detection in liver, before been processed for staining, 8-μm sections of OCT preserved frozen liver tissue were fixed with 4% PFA for 15 min. Fluorescence microscopy was performed using an inverted microscope (Zeiss Apotome 2) using 40× oil-immersion objective. The coded slides were examined in a blinded fashion. For histochemical analyses, liver tissue was fixed overnight in phosphate-buffered 10% formalin and embedded in paraffin. A total of 6-μm sections were cut and processed for staining with HE.

**Proximity ligation assay.** Proximity ligation assays were performed on HEK293T cells (acquired from ATCC, bi-weekly tested for mycoplasma-free status) transfected either with empty vector (control cells) or plasmids expressing PPARα-HA with or without Vps15-flag. The vector expressing GFP protein was co-transfected in all conditions to quantify the interactions in transfected cells. The assays were carried out according to the manufacturer's instructions (DUO92101 - Duolink® In Situ Red Starter Kit Mouse/Rabbit, Sigma). In brief, transfected cells were fixed with 4% paraformaldehyde for 20 min and permeabilized with 0.1% Triton X-100 in PBS for 20 min at room temperature followed by blocking for 30 min at 37 °C. The cells were incubated with primary antibodies to Hdac3 (1:200, Cell Signalling) and to HA-tag (1:200, Sigma) for 1 h at 37 °C. The incubation with

PLA probe PLUS and MINUS conjugated with oligonucleotides was performed for 1 h at 37 °C. The terminal steps of ligation and amplification were performed at 37 °C for 30 and 90 min, respectively. Images were acquired using an inverted microscope (Zeiss Apotome 2) using 60× oil-immersion objective.

**Protein extraction, immunoblotting and immunoprecipitation.** To prepare protein extract for immunoblot analysis, cells were washed twice with cold phosphate-buffered saline (PBS), scraped from the dishes in lysis buffer containing 20 mM Tris-HCl (pH 8.0), 5% glycerol, 138 mM NaCl, 2.7 mM KCl, 1% NP-40, 20 mM NaF, 5 mM EDTA, 1× protease inhibitors (Roche), 1× PhosphoStop Inhibitors (Roche). The same buffer was used to prepare protein extracts from liver tissue. Homogenates were spun at $12,000 \times g$ for 10 min at 4 °C. For immunoprecipitation 500 μg of cleared protein extract was incubated with 1 μg of anti-HA (Sigma) antibody for 3 h at +4 °C. Then, immune complexes were pulled down using Protein G Sepharose beads (GE) during 2 h followed by four washes with extraction buffer. The protein complexes were eluted by boiling the beads in 1xSDS-sample buffer for 10 min. Protein extracts or immunoprecipitate eluates were resolved by SDS-PAGE before transfer onto PVDF membrane followed by incubation with primary antibodies and HRP-linked secondary antibodies. Protein extracts were resolved by SDS-PAGE before transfer onto PVDF membrane followed by incubation with the primary antibodies and HRP-linked secondary antibodies. Immobilon Western Chemiluminescent HRP Substrate (Millipore) was used for the detection. The images were acquired on ChemiDocTM Imager (Biorad). The uncropped images of blots could be found as Supplementary Information.

**GFP-trap assay.** For immunoprecipitation with GFP-LC3 and GFP-GABARAP proteins, HEK293T cells were transiently transfected with plasmids expressing GFP, GFP-LC3B and GFP-GABARAP proteins using JetPei reagent (Thermo). Twenty-four hours post transfection, cells were collected and proteins extracted in lysis buffer (10 mM Tris, pH 7.5, 150 mM NaCl, 0.5 mM EDTA, 0.5% NP-40) complemented with protease and phosphatase inhibitor cocktail (Pierce). Cell lysates were centrifuged at $15,000 \times g$ for 10 min at +4 °C. The resulting supernatant was diluted with the dilution buffer (10 mM Tris, pH 7.5, 150 mM NaCl, 0.5 mM EDTA) to NP-40 final concentration of 0.1%. The protein extracts were incubated with anti-GFP beads (GFP-Trap Chromotek) for 1 h at +4 °C. Beads were collected by centrifugation and washed six times, the protein complexes were eluted by boiling the beads in 1×SDS-sample buffer for 10 min.

**Real-time quantitative PCR.** Total RNA was isolated from liver tissue using RNAeasy Lipid Tissue Mini Kit (Qiagen) and RNeasy Mini Kit (Qiagen) from primary hepatocytes. Single-strand complementary DNA was synthesized from 1 μg of total RNA using 125 ng of random hexamer primers and SuperScript II (Life Technologies). RT-qPCR was performed on MX3005P instrument (Agilent) using a Brilliant III Ultra-Fast QPCR Master Mix (Agilent). The relative amounts of the mRNAs studied were determined by means of the $2^{-\Delta\Delta CT}$ method, with geometric mean of pinin, S18, cyclophilin, eIF2α, HUS, Ubiquitin as reference genes and control treatment or control genotype as the invariant control. The primer sequences are listed in Supplementary Table 1. For mtDNA quantification, total DNA from liver tissue was isolated using DNAeasy Blood and Tissue kit (Qiagen) according to the manufacturer's instructions.

**Chromatin immunoprecipitation.** Chromatin immunoprecipitation was performed with anti-PPARα antibody (Santa Cruz, sc-398394) using primary hepatocytes or anti-H3K27Ac antibody (Abcam, ab4729) using liver tissue as described previously[65]. The relative amount of the immunoprecipitated DNA was determined by RT-qPCR using the $2^{-\Delta\Delta CT}$ method, with input DNA values for each sample as control. The primer sequences are listed in Supplementary Table 1.

**Gene expression profiling and bioinformatic analyses.** Total RNA was isolated from liver tissue of Vps15$^{f/f}$ mice transduced with adenoviral vectors expressing GFP or GFP-Cre protein using RNAeasy Lipid Tissue Mini Kit (Qiagen) according to manufacturer's protocol. Complementary RNA was synthesized, amplified and purified using the Illumina TotalPrep RNA Amplification Kit (Ambion) following the manufacturer's recommendations. Briefly, 200 ng of RNA was reverse transcribed. After second-strand synthesis, the complementary DNA was transcribed in vitro and complementary RNA labeled with biotin-16-UTP. Labeled probe hybridization to Illumina BeadChips Mouse WG-6 v2 (Illumina) was carried out using Illumina's protocol. The Beadchips were scanned on the Illumina IScan using Illumina IScan image data acquisition software. Illumina GenomeStudio software (Illumina) was used for preliminary data analysis. The Illumina data were then normalized using the 'normalize quantiles' function in the GenomeStudio Software (Illumina). We used a threshold at 0.01 to convert 'Detection pval' into flags: flag = 0 if pval > 0.01 and flag = 1 if pval <= 0.01. For changes at the transcriptional level, total RNAs from livers of two different conditions were compared. The group comparisons were done using Student's $t$-test on probes flagged as '1' for at least half of the samples involved in the comparison. We filtered the resulting P-values at 5% and fold change 1.5. Heat map representations of log2 normalized expression data were generated using R (http://www.r-project.org/foundation/) and Treeview software (http://taxonomy.zoology.gla.ac.uk/rod/treeview.html) using the

Spearman correlation similarity measure and average linkage algorithm. The gene list of all significantly modified genes (filtered fold change 1.5) could be found in Supplementary Data 1. GO analyses were performed using the Database for Annotation, Visualization and Integrated Discovery (DAVID) V6.8. The gene list was generated by selecting significantly modified genes in liver samples of Vps15f/f mice transduced with Adeno-Cre viral vectors (depleted of Vps15). For GO term analysis, the biological process, cellular component and molecular function categories were studied using the GO FAT default settings. Functional annotation clustering was performed with the default criteria at medium classification stringency.

**Statistical analysis**. Data are shown as means ± SEM. The unpaired two-tailed Student's $t$-test was applied for statistical analysis. Results were considered significant in all experiments at $P < 0.05$.

**Reporting summary**. Further information on experimental design is available in the Nature Research Reporting Summary linked to this article.

## Data availability

Material and Correspondence: All data are available from the corresponding author upon reasonable request. The information and requests for resources and materials should be directed to Dr. Ganna Panasyuk (ganna.panasyuk@inserm.fr). The source data underlying all figures in the main text of the manuscript are provided as a Source Data file. The source data for microarray analyses in Vps15 depleted liver are deposited in ArrayExpress database E-MTAB-7685.

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

## Acknowledgements

We are grateful to all lab members and the members of INSERM-U1151 for support, to Patrice Codogno for helpful discussions and Nathaniel Henneman and Nicolas Kuperwasser for manuscript editing; Bertrand Blondeau for Ad-PGC1α; Maurizio Crestani for sharing Ad-shRNA Hdac3; Philippe Noirez and Marion Falabregue for help with DEXA scan measurements; Sophie Berissi (SFR Small animal histology and morphology platform) and Sylvie Fabrega (SFR Viral vector and gene transfer platform) for technical support. This work was supported by grant from Agence National de la Recherche (ANR) to G.P. (ANR-JCJC-NUTRISENSPIK-16-CE14-0029) and European Research Council (ERC) to M.P. The research leading to these results has received funding from People Programme (Marie Curie Actions) of the European Union's Seventh Framework Programme (FP7/2007–2013) under REA grant agreement no. PCOFUND-GA-2013–609102, through the PRESTIGE programme coordinated by Campus France. A.I. was supported by ANR-JCJC-NUTRISENSPIK-16-CE14-0029 and PRESTIGE programme. C.A. was supported by Boulos Foundation. H.G. was supported by Hepatokind grant from ANR. E.I.R was supported by Deutsche Forschungsgemeinschaft (SFB1218/A05).

## Author contributions

A.I. and I.N. conducted most of the experiments, analysed the data, prepared the figures and contributed to the manuscript writing. C.A. performed animal experiments and interaction studies in cells and participated in data analyses. M.G. performed histological examination of liver tissue samples. N.C. performed the bioinformatic analyses of microarrays. E.R. and E.B. performed electron microscopy analyses. D.C. measured activity of respiratory chain complexes. H.G. and A.M. provided the PPARα mutants, shared reagents and expertise. G.P. and M.P. conceived the study and obtained funding. G.P. directed the work, designed the experiments, conducted experiments, analysed the data and wrote the manuscript. All authors discussed the results and commented on the manuscript.

## Additional information

**Competing interests:** The authors declare no competing interests.

