## [Peer Review File · Nature Communications]

Point-by point response

We thank the reviewers for their helpful comments and we appreciate they found specific interests in our work. In this revised version, we provide **requested additional controls** as well as a substantial amount of **new functional data** addressing the mechanistic links between the class 3 PI3K and lysosomal degradation of PPAR α repressors. Briefly, we address **three main points**:

1) We now provide a large amount of new functional data on regulation of NCoR1 and Hdac3 by class 3 PI3K. In addition to Hdac3, in this revised version of the manuscript, we show that Hdac1 is also accumulated in liver of Vps15 mutant. Importantly, we now show that NCoR1 and Hdac3 are degraded by autophagy *in vivo* in liver. Significantly, we provide mechanistic insights into the selective autophagy of NCoR1 and Hdac3 stimulated by the class 3 PI3K. We now show that NCoR1 and Hdac3 interact with Atg8-like proteins GABARAP and LC3. Furthermore, we find that this interaction is modulated by Vps15 overexpression and by pharmacologic inhibition of class 3 PI3K. Altogether, these mechanistic analyses considerably reinforce our initial conclusion on the implication of class 3 PI3K in the control of PPAR α transcriptional responses in liver.

2) We further strengthened our initial findings on the mitochondrial dysfunction in Vps15-null models. Thanks to reviewer guidance, we now show that mitochondrial respiration and mitochondrial DNA content are significantly decreased in Vps15-null liver. In addition, we have demonstrated that the fasting-induced increase in mitochondrial mass is Vps15 and PPAR α dependent. We believe that these novel findings further support our important discovery that PPAR α transcriptionally controls mitochondrial biogenesis in liver.

3) We complemented the analyses in the Vps15 mouse model and in cells with all requested controls. Essentially, in this revised manuscript, we have used genetic tools to downregulate Hdac3 and Vps34, the catalytic subunit of the class 3 PI3K, to demonstrate that those are sufficient to significantly restore or reproduce the mitochondrial phenotype of Vps15-null models, respectively. Furthermore, as guided by the reviewer suggestions, we performed *in vivo* analyses of the adipose tissue and biochemical analyses of plasma in Vps15 hepatic mutant and now report, for the first time in this model, a complex phenotype of increased lipolysis and defective whole-body lipid metabolism.

There is a large amount of new data that are included in the **7 main figures, and 7 supplementary figures** of this revised manuscript which support the novel function of class 3 PI3K upstream of PPAR α and lipid degradation in liver. Below is our detailed response to the reviewer's comments. The modifications to text of the manuscript are **highlighted in red**.

Reviewer 1:

We would like to thank the reviewer for highlighting the novelty and importance of our study towards a better understanding the mechanisms of metabolic adaptation in physiological fasting and the novel role of class 3 PI3K in control of PPAR α transcriptional activity.

The present work analysed the involvement of VPS15 and autophagy in lipid metabolism. The authors performed transcriptomic analyses of livers in which VPS15 was acutely deleted and compared with controls. The data suggest that inhibition of VPS15 affected pathways related to mitochondria and lipid homeostasis. Metabolomic analyses on liver-specific VPS15 knockout mice during fasting confirmed the changes in lipids and underlined an accumulation of carnitine-conjugated suggesting a problem in lipid oxidation. Because PPAR α is an important player in lipid metabolism in liver they compared the transcriptomic profiles of PPAR α KO and VPS15 KO mice and found strong similarities. When they checked the nuclear localization of PPAR α and the repressor NCoR1/HDAC3 complex and found a decreased and increased nuclear localization, respectively. By using fenofibrate or PGC1 overexpression to reactivate PPAR α the authors showed a partial rescue of lipid metabolism. The message that VPS15 controls lipids via PPAR α is novel and interesting but the insights of this regulation are weak. It is not clear whether this is a direct consequence of autophagy inhibition or not and whether autophagy control NCoR1/HDAC3 or something else that impinge on PGC1a/PPAR α axis. No genetic approach has been used to address the role of NCoR1/HDAC3 axis on phenotype of VPS15 KO mice. The authors should consider the following points.

Point 1. The authors used AAV to express Cre-recombinase in muscles and in cells that have the VPS15 floxed gene. However, overexpression of Cre is known to generate artefacts and this is the reason why Cre-transgenic mice that are crossed with floxed mice are always kept in het condition and never in homozygous for Cre. Authors must check whether the transcriptomic profile of AAV-mediated overexpression of Cre in skeletal muscle of wild type mice is affected and test the changes of mitochondrial function and morphology in vitro after cre overexpression in control cells. This is a critical control experiment for the correct interpretation of the data. Authors must also show the level of deletion and expression of VPS15 after AAV-Cre expression and show the morphology of the muscles after viral infection. It is well known that AAV may induce myopathy per se when injected in high doses.

Response:

We agree with the reviewer that special precautions have to be taken when experimenting with virally expressed recombinant Cre protein. Indeed, we have made those controls on wild-type mice before proceeding with experiments on cells from Vps15 floxed mice and when expressing Cre *in vivo* in livers of Vps15 floxed mice. We express Cre protein using non-replicative adenoviral vectors (Ad5-Cre-GFP) in the lowest dose in which efficient deletion of Vps15 is achieved without any noticeable effect on cell metabolic activity and viability. Importantly, we always use a control infection with the adenoviral vectors expressing GFP protein (Cre protein that we use is a GFP-fusion). As an example, we present in this revised manuscript that Cre expression in primary hepatocytes isolated from wild type mice does not affect mitochondrial respiration assessed by Seahorse assay or the expression of

PPAR α targets. These new data are presented in revised manuscript as **new Supplementary Fig. 2e, 2f**.

Notably, our work is entirely focused on liver and we have used all generally accepted controls to assure adequate interpretation of the phenotypes (transduction with control viral vectors; Alb-Cre transgene always kept in Het state in mice; analyses of Alb-Cre negative and Alb-Cre positive littermates). We employed a combination of acute (shRNA, Ad5-Cre-GFP) and chronic (Alb-Cre) genetic depletion approaches and pharmacologic (Vps34 inhibitor) approaches. These approaches are complementary and the results support the hypothesis that functional class 3 PI3K signalling is required for transcriptional activity of PPAR α .

Point 2. Figure 1d. Respiration experiment is problematic. In fact, it looks like that the ATP-dependent respiration (the difference between the basal respiration and after oligomycin treatment) is less affected by VPS15 knockdown when compared to control and is unaffected by AAV-Cre mediated VPS15 ablation. Authors must include wild type cell transfected by AAV-cre. These data suggest that mitochondria can generate ATP in a basal condition and that the reduced respiration is consequent to a mild uncoupled condition that explains the reduced maximal respiration after FCCP. Authors must check respiratory complex activity, mitochondrial membrane potential and ROS generation before claiming that mitochondria are dysfunctional. Moreover, a better characterization of mitochondrial network should be performed (EM is insufficient to explore the network, see below) including mtDNA content and mitochondrial mass quantification.

Response:

We would like to thank the reviewer for these insightful suggestions that allowed us to reinforce our main conclusions on mitochondrial dysfunction in Vps15 mutant. As suggested by all reviewers, in this revised version of the manuscript, we provide new data on the biochemical profiling of the respiratory chain complex activities in liver extracts of control and Vps15 mutants. We revealed a significant inhibition of the total respiratory chain activity due to decreased activity of all five complexes in liver tissue of Vps15 mutant (**new Fig. 1d, new Supplementary Fig. 2c**). Notably, this defective respiration in Vps15 mutants was also accompanied by significant increase in lactate dehydrogenase activity in livers of Vps15 mutant (**new Supplementary Fig. 2d**).

Also, as suggested by the reviewer, we now show that respiration is not affected in wild type hepatocytes transduced with adenoviral vectors expressing Cre recombinase (**new Supplementary Fig. 2e**). In addition, we now demonstrate that not only inactivation of Vps15 but also knockdown of Vps34, a lipid kinase subunit in class 3 PI3K complex, inhibit mitochondrial respiration (**new Supplementary Fig. 6a**). As suggested by reviewer, we have also improved the presentation of Seahorse measurements and now show, for each experiment, the graphs of spare respiratory capacity and ATP production with the statistical analyses conducted (**new Fig. 1e, 5a, 6g and Supplementary Fig. 6a**). Finally, we also performed staining with the potential sensitive dye MitoTracker Red in primary hepatocytes

isolated from wild-type and Vps15-LKO mice (**Figure for Reviewer 1a**). This staining revealed weak signal which is consistent with a significant mitochondrial depletion in Vps15-null hepatocytes. To this end, as suggested by the reviewer, we also measured mtDNA content in liver tissue of Vps15 hepatic mutant. We now show that, consistent with decreased mitochondrial mass, mtDNA is depleted in liver of Vps15 mutant (**new Supplementary Fig. 2b**). In addition, in this revised manuscript we show that fasting-induced increase in mtDNA content in liver tissue of control mice requires expression of Vps15 (**new Fig. 7c**). Furthermore, we now show that this response is also dependent on PPAR α , as it is fully abrogated in fasted hepatic mutant of PPAR α (**new Fig. 7d**).

Figure for Reviewer 1. Mitochondrial dysfunction in Vps15-null hepatocytes. **a** Mitochondria depletion in Vps15-null hepatocytes. Isolated primary hepatocytes from Vps15^{ff/ff} and AlbCre⁺;Vps15^{ff/ff} mice were treated with 50nM MitoTracker Red for 30 min and PFA fixed. Images acquired with inverted fluorescent microscope Zeiss Apotome 2. Scale bar: 40 μ m. **b** Increased oxidative stress in Vps15-depleted cells. Isolated from Vps15^{ff/ff} mice primary hepatocytes were infected with adenoviral vectors expressing GFP, Cre or shRNA Vps15. Cells were collected 72 hours post-infection for metabolic measurements by mass spectrometry. GSH:GSSG ratio are represented as fold changes over GFP-infected hepatocytes. Data are means \pm SEM ($P < 0.05$ *: vs GFP-infected cells). **c** Carbonylated proteins in livers of six week old random-fed Vps15^{ff/ff} and AlbCre⁺;Vps15^{ff/ff} mice were labeled with specific functionalized fluorescent probes and samples were resolved high-resolution electrophoresis separation (OxiProteomics). Image acquisition for carbonylated proteins was performed using the Ettan[®] DIGE imager (GE Healthcare).

Moreover, as advised by the reviewer, we complement the immunofluorescent analyses of the mitochondrial network in Vps15 depleted hepatocytes by performing additional staining with Tom40, an integral protein of mitochondrial membrane (**new Fig. 1c**). These immunofluorescent analyses show that Tom40-positive mitochondrial compartment is disturbed (less mitochondria and fragmented network) in Vps15-null hepatocytes. This is consistent with our new data on the activated fission in livers of Vps15-LKO mice (**new Supplementary Fig. 7a**). Altogether, these findings are consistent with data presented in the previous version of the manuscript (decreased staining of cytochrome c, decreased

mitochondrial biogenesis and fragmented mitochondrial network revealed by ultrastructural analyses) and our new data on mtDNA depletion in livers of Vps15 mutant.

Finally, as suggested by reviewer, we investigated putative oxidative stress in Vps15-depleted cells. Additional bioinformatic analyses of the microarray datasets of liver tissue upon acute Vps15 depletion, that we have now performed, revealed the “Disulfide bond” as the top pathway among upregulated genes (**new Supplementary Fig. 1a, new Supplementary Data 2**). Importantly, by performing transcript and protein expression analyses in liver tissue of Vps15 mutants, we have confirmed that the signature of oxidative stress response is highly upregulated upon Vps15 inactivation (**new Supplementary Fig. 1**). It is evidenced by significant overexpression of GST, *Nqo1*, *Gpx7* and *Gstm2* both at protein and transcript level (**new Supplementary Fig. 1c, 1d**). This is fully consistent with increased protein expression of Nrf2 transcription factor that we have observed in livers of Vps15 hepatic mutants (**new Supplementary Fig. 1b**). In line with these findings in livers of Vps15-LKO mice, we also conducted additional metabolomic analyses in primary hepatocytes which revealed that GSH:GSSG ratio is highly downregulated in primary hepatocytes depleted of Vps15 further advocating on-going oxidative stress in Vps15-null cells (**Figure for Reviewer 1b**). Finally, analyses of protein carbonylation in livers of Vps15-LKO mice suggested pattern of increased protein oxidation in liver tissue of Vps15 mutant compared to wild-type mice (**Figure for Reviewer 1c**). In sum, we believe that these extensive new analyses, combined to the previous transcriptomic, metabolomic and ultrastructural findings, firmly support our conclusion on the mitochondrial dysfunction upon Vps15 inactivation.

Point 3. It is unclear how VPS15 KO mice can survive in fasting condition. The authors claimed that are hypoglycemic and hypoketogenic. But ketons are required to cardiac and brain function when glucose is unavailable/reduced such as in 24-fasting or when insulin is lacking (Type1 diabetes). In absent to ketones and glucose brain would not have energy and mice should die. How the authors explain this metabolic paradox? How is the lactate level ins the blood after fasting?

Response:

We thank the reviewer for raising several important points on the complex metabolic phenotype of liver-specific mutant of Vps15. This led us to further highlight the importance of hepatic class 3 PI3K signalling for whole-body metabolic homeostasis. Notably, our findings in liver-specific Vps15 knockout mice are consistent with the metabolic phenotype of a whole-body and liver specific PPAR α knockout mice (Nemazanyy, et al. 2015), (Leone, et al. 1999), (Montagner, et al. 2016). The PPAR α mutants are viable, however, when challenged with starvation, are hypoketogenic and hypoglycaemic. We agree with the reviewer that most likely the metabolic plasticity of these mutants is limited under starvation stress and, if fasting were prolonged, would be detrimental to mutant viability. We have challenged both PPAR α and Vps15 mutants by fasting for 24 hours, the longest duration allowed by our ethics permit, and

observed a significant decrease in the metabolic responses such as ketone body production. Moreover, the liver is not the only site of ketogenesis and gluconeogenesis, as the kidneys and intestines are also reported to contribute (Takagi, et al. 2016), (Owen, et al. 1969), (Zhang, et al. 2011), (Bekesi and Williamson 1990), (Penhoat, et al. 2014). Their contribution in fasting induced ketogenesis in hepatic mutant of Vps15 is out of the scope of this work. As suggested by reviewer, we have also measured the lactate levels in fasted and fed mice (**new Supplementary Fig. 3f**). These analyses revealed that plasmatic lactate is non-significantly decreased (p value 0.059) in fed Vps15 mutants with fasting decreased circulating lactate to a comparable level in control and Vps15 mutants. These findings suggest that lactate is not an alternative energetic source in Vps15 mutants after prolonged fast. Our future work will determine the metabolic processes that are induced in the liver tissue of Vps15 hepatic mutant to compensate the lack of PPAR α transcriptional activity.

Another metabolic paradox is related to lipid metabolism. During fasting there is an important lipolysis that induces a liver overload causing liver steatosis. Liver steatosis should be enhanced in case of mitochondrial beta oxidation inhibition due to CPT1 reduction. Instead authors found a decrease of FFA or triglyceride in liver of VPS15 KO mice. The only explanation is that mitochondrial are uncoupled and even in face of a reduced CPT1/CPT2 expression are still able to greatly use acyl-CoA for beta-oxidation. Authors must check the blood levels of FFA in fasting and the expression of UCP proteins in liver. Finally, quantification of WAT in VPS15 KO mice in this experimental setting must be also performed.

Point 6. The data on triglyceride in plasma or liver suggest that there is a problem in lipid uptake. Is it the case?

Response:

The reviewer is absolutely right that fasting induces the lipolysis in adipose tissue and provokes an influx of lipids in liver used as a substrate for the ketogenesis and β -oxidation. To assess the changes in adipose tissue in response to fasting, we have performed the benchmark measurements of adipose tissue content using dual energy X-ray absorptiometry (DEXA) scan in control and Vps15 hepatic mutants in fed and fasted states. Three major conclusions could be drawn from these new data presented as **new Supplementary Fig. 3g-j**. First, hepatic mutants of Vps15 present significantly lower adiposity both in fed and fasted state (**new Supplementary Fig. 3g**). Second, this decreased adiposity is paralleled by significantly increased lean mass in Vps15 liver-specific mutant mice (**new Supplementary Fig. 3h**). Finally, despite significantly decreased adiposity of Vps15 liver mutants, the fasting induced loss of fat in these mice was higher as compared to wild type mice (**new Supplementary Fig. 3j**). The body fluid content was unmodified in fed/fasted wild type and Vps15 mutant mice (**new Supplementary Fig. 3i**). In sum, these findings suggest that hepatic class 3 PI3K signalling has a whole-body effect on adipose tissue maintenance in response

to fasting and, potentially, acts as a brake of the fasting induced lipolysis. The mechanistic studies in this direction are out of scope of current manuscript and will be pursued in future.

Following the suggestion of all three reviewers, we have also performed extensive biochemical studies in plasma of Vps15 hepatic mutants in fed and fasted state. These new data are presented as **new Supplementary Fig. 3f**. These analyses have revealed that, as expected, fasting in control mice resulted in increased plasmatic levels of glycerol and free fatty acids (**new Supplementary Fig. 3f**). Moreover, fasting led to a decrease in plasmatic TG levels consistent with lipid uptake by liver, as also witnessed by hepatic steatosis in wild type mice (**new Supplementary Fig. 3b-3d, 3f**). At the same time, in Vps15 hepatic mutants, the levels of free fatty acids and TG were significantly increased in fed state and TG levels were unmodified by 24 hour fasting (**new Supplementary Fig. 3f**). Notably, consistent with changes in fat mass measured by DEXA scan, the fasting of Vps15 hepatic mutants resulted in further increase in plasmatic levels of free fatty acids (**new Supplementary Fig. 3f**). It was also paralleled by significantly lower levels of circulating glycerol both in fed and fasted state (**new Supplementary Fig. 3f**). In sum, these findings demonstrated that plasmatic lipid homeostasis is defective in Vps15 hepatic mutants. These observations advocate that lipid uptake is controlled by class 3 PI3K in hepatocytes.

[redacted]

Although we cannot exclude that the uncoupling in mitochondria takes place in Vps15-null hepatocytes, we believe extensive data presented in this revised manuscript strongly advocates a positive role of Vps15 in control of PPAR α transcription for mitochondrial biogenesis and fatty acid degradation. To this end, an increased ratio of acyl-carnitine

[redacted]

derivatives to free carnitine is an indicator of impaired fatty acid oxidation. We present now on **new Fig. 2g** and **new Supplementary Fig. 3k** that while the levels of free carnitine are

comparable in liver tissue of fed mice, they are significantly decreased in livers of fasted Vps15 mutant compared to fasted control mice. Altogether, mitochondrial mass reduction, inhibition of the respiratory chain activity, decreased expression of fatty acid trafficking proteins, significant accumulation of long chain fatty acid acyl-carnitines together with lower levels of acetyl-CoA and free carnitine support our initial hypothesis on defective fatty acid degradation in Vps15-null hepatocytes.

Point4. Liver morphology was never shown and instead must be displayed in the different KO mice and conditions (AAV-mediated Cre expression; fed and 24 hours fasting).

Response:

We thank the reviewer for raising this point that improved the illustration of our findings. We have now included the images of gross liver morphology and HE stained liver sections of fed and fasted WT and AlbCre⁺;Vps15^{ff} mice (**new Supplementary Fig. 3b, 3c**). We have already published the images of gross liver morphology and HE stained liver sections of Vps15^{ff} mice transduced with Ad5-Cre-GFP vectors as well as AlbCre⁺;Vps15^{ff} mice (Nemazanyy et al., Nat Comm. 2015). Those are also presented on **Figure for Reviewer 3**. We have added this information when describing the model and mentioned that, in these models, Vps15-depletion concurs with autophagy block. In addition, we present new data clearly showing that while fasting induces a significant decrease in liver size in control mice, the Vps15 hepatic mutants are resistant to this physiological response (**new Supplementary Fig. 3e**).

Point 5. How is the mRNA expression of PPAR α in VPS15 KO Mice? Is the PPAR α protein turnover increased?

Response:

We thank reviewer for raising this important point that helped us to understand better the regulation of PPAR α downstream of Vps15. The lack of activating ligand and the presence of co-repressors activate proteosomal degradation of PPAR α . Notably, we have demonstrated that fenofibrate treatment significantly rescued nuclear levels of PPAR α (**new Supplementary Fig. 4b**). We now show that PPAR α transcript (**new Supplementary Fig. 3a** and **new Supplementary Fig. 4h**) and protein (**Fig. 2f** and **new Supplementary Fig. 4g**) are downregulated in livers of Vps15 hepatic mutants. As for protein turnover, we discovered that PPAR α protein ubiquitination is increased in livers of Vps15 hepatic mutants (**Figure for Reviewer 4**). We used in this experiment Tandem Ubiquitin Binding Entity beads that bind with equal affinity K63 and K48 ubiquitinated proteins (Tube2). As presented on **Figure for Reviewer 4a**, despite profound PPAR α protein depletion in the livers of Vps15 mutant mice,

Figure for Reviewer 3. Characterisation of mouse models of Vps15 deletion in liver. **a** Immunoblot analyses of liver extracts of random-fed Vps15^{ff} mice sacrificed at different times post-transduction with Adeno-GFP or Adeno-CRE vectors using indicated antibodies. **b** Relative mRNA expression levels of class 3 PI3K complex subunits in the livers random-fed mice ten days post-transduction with Adeno-GFP or Adeno-CRE vectors. Data are means ±SEM (n=5-8, P<0.05 *: vs Adeno-GFP, 2-tailed, unpaired Student's t test). **c** Macroscopic view of livers of random-fed mice ten days post-transduction with Adeno-GFP or Adeno-CRE vectors. **d** Liver sections of random-fed mice were immunostained with anti-BrdU and anti-β-catenin antibodies to label proliferating cells (BrdU incorporation) and to facilitate morphometric analysis by labelling plasma membrane (β-catenin) (left panel). **e** Immunoblot analysis of protein expression in livers of one month old random-fed Vps15^{ff} and AlbCre⁺;Vps15^{ff} mice with indicated antibodies. Probing with anti-GAPDH antibody served as loading control. **f** Relative mRNA expression levels of class 3 PI3K complex genes in the livers of one month old random-fed Vps15^{ff} and AlbCre⁺;Vps15^{ff} mice. The recombination occurs in exon 2 of Vps15 gene locus. The analyses using primers nested in exon 4 show lack of compensatory overexpression of truncated Vps15 product. Data are means ±SEM (n=5-8, P<0.05 *: vs Vps15^{ff}, 2-tailed, unpaired Student's t test). **g** Macroscopic view of livers of one month old random-fed Vps15^{ff} and AlbCre⁺;Vps15^{ff} mice. **h** HE stained liver sections of one month old random-fed Vps15^{ff} and AlbCre⁺;Vps15^{ff} mice. Scale bar: 50 μm.

similar levels of PPARα protein are pulled down on Tube2 beads from total protein extracts of control and Vps15-null liver. These observations are paralleled by our findings of increased

protein expression of Huwe1 protein, an E3 ubiquitin ligase that was recently shown to control polyubiquitination of PPAR α in hepatocytes (**Figure for Reviewer 4b**)(Zhao, et al. 2018). We have also demonstrated that Huwe1 is activated in livers of Vps15 mutants, as evidenced by its increased binding to Tube 1 beads which have higher affinity to K63 ubiquitinated proteins, a common posttranslational modification for active E3 ligases (**Figure for Reviewer 4c**). Finally, we show that PPAR α protein turnover is modulated by Vps15 expression. To this end, as shown in the **new Supplementary Fig. 5b**, Vps15 overexpression in HEK293T cells increased basal levels of recombinant PPAR α protein and slowed its degradation, as monitored upon cycloheximide addition. These are in line with our new findings presented in **new Fig. 4d and 4e**, that Vps15 overexpression decreases complex of PPAR α with Hdac3. In sum, these new findings are concordant with the positive role of PPAR α in control of its own transcription and, also, they are in agreement with known mechanism of PPAR α proteosomal degradation. We believe that, although these new data are interesting, they are out of the main focus of the current report and, if reviewer agrees, they will not be included in the manuscript to avoid distraction.

Figure for Reviewer 4. Ubiquitination of PPAR α is induced in livers of Vps15-LKO mice. **a** Endogenous PPAR α and Huwe1 proteins were pulled-down on Tube2 agarose from 1mg of total liver extract of six week old random-fed Vps15^{f/f} and AlbCre⁺;Vps15^{f/f}. The precipitation with control agarose beads served as a control of non-specific binding. Immunoblot analyses with indicated antibodies revealed similar pattern of binding of analysed proteins precipitated from liver tissue extracts of wild-type and Vps15-LKO mice. **b** Immunoblot analysis of total protein liver extracts of random-fed six week old Vps15^{f/f} and AlbCre⁺;Vps15^{f/f} using indicated antibodies. Immunoblot with GAPDH antibody served as a loading control. **c** Endogenous Huwe1 protein was pulled-down on Tube1 agarose from 1mg of total liver extract of six week old random-fed Vps15^{f/f} and AlbCre⁺;Vps15^{f/f}. The precipitation with control agarose beads served as a control of non-specific binding. Immunoblot analyses with anti-Huwe1 antibody revealed increased binding of Huwe1 (K63-Ubiquitination) in liver tissue extracts of Vps15-LKO mice.

Point 7. It is not clear how the changes of NCoR1 HDAC3 proteins are due. The fig 3e suggest that are consequent to autophagy but more data are required. Is it mediated by p62 or do they contain LIR domains and are directly delivered to autophagosomes? How they can get out of the nucleus and incorporated into the autophagosomes (Suppl Fig 2d)? Is PPAR α protein affected by lysosomal inhibition? In case that lysosomes control the NCoR1 and HDAC3 degradation how can fenofibrate reduce their levels (Fig 3c) in VPS15 KO mice? Western blots of total protein are required to quantify NCoR1 and HDAC3 level in presence and absence of fenofibrate.

Response:

We completely agree with the reviewer that additional mechanistic insights would further deepen our understanding of the role of Vps15 and class 3 PI3K in control of NCoR1 and Hdac3 proteins autophagic degradation. A similar point was also raised in the comments of two other reviewers. Following the insightful comments of all reviewers, we approached this question by further studying the *in vivo* autophagic degradation and the mechanisms of selective autophagy of these transcriptional repressors. First, we show that Hdac3 and NCoR1 proteins are degraded in lysosomes upon starvation-induced autophagy. It is evidenced by their accumulation in livers of fasted wild type mice treated with Leupeptin inhibitor (lysosomal protease inhibitor) (**new Fig. 3e, new Supplementary Fig. 4e**). Notably, this degradation is abrogated in Vps15-null livers, consistent with defective autophagic flux in mutants (**new Fig. 3e, new Supplementary Fig. 4e**). This is consistent with our findings of Hdac3 and NCoR1 accumulation in primary hepatocytes treated with Bafilomycin A1 (**new Fig. 3f**). In addition, in this revised manuscript, we show that endogenous Hdac3 and NCoR1 proteins co-localize with Lamp2 protein in primary hepatocytes (**Fig. 4a**) further supporting our conclusion on their lysosomal degradation.

Second, guided by the reviewer, we tested if Hdac3 and NCoR1 proteins could be targets of selective autophagy. In this revised manuscript, we now show that, Hdac3 and NCoR1 proteins interact with GABARAP and LC3B proteins (**new Fig. 4b**). Notably, the interaction of NCoR1 was observed predominantly with GABARAP while Hdac3 was pulled down with GABARAP and to lesser extent with LC3B, suggesting distinct mechanisms for the induction of their selective autophagy. Our further studies have demonstrated that interaction of Hdac3 and NCoR1 with GABARAP protein was increased by Vps15 overexpression (**new Fig. 4c**) and could be reduced by pharmacologic inhibition of class 3 PI3K with the selective inhibitor PIK-III (**new Fig. 4b**).

Third, we now show that in line with increased interaction of Hdac3 protein with GABARAP, Vps15 overexpression promoted dissociation of Hdac3 from PPAR α protein. This is demonstrated by decreased Hdac3 levels in PPAR α immunoprecipitates (**new Fig. 4d**) and significantly downregulated formation of complexes between Hdac3 and PPAR α , as revealed by proximity ligation assay (**new Fig. 4e**). Altogether, these novel findings strongly advocates that activated class 3 PI3K signalling promotes selective autophagy of PPAR α transcriptional repressors.

In this point, the reviewer also raised the question about possible lysosomal degradation of PPAR α . We tested it and, consistent with published reports on proteosomal degradation of PPAR α , its protein levels are unmodified by lysosomal inhibition in *ex vivo* explants of liver tissue, unlike the levels of NCoR1 repressor, Lamp2 and LC3-II proteins (**Figure for Reviewer 5**).

Figure for Reviewer 5. PPAR α is not targeted for lysosomal degradation. Immunoblot analysis of total protein liver extracts of *ex vivo* autophagic flux experiment in six-week old wild type mice using indicated antibodies. The livers were perfused with PBS solution, dissected and minced. The tissue was incubated during four hours at +37°C in EBSS solution supplemented with Chloroquine at concentration 100 μ M or 500 μ M. Tissue was collected by brief centrifugation and washed once with PBS before proceeding with protein extraction. Immunoblot with anti-Lamp2 and anti-LC3 antibodies served as a control of lysosomal inhibition (accumulation of Lamp2 and LC3-II protein) and anti-GAPDH as a loading control.

Finally, as requested by the reviewer, we performed whole liver protein expression analyses after fenofibrate treatment. These analyses are consistent with immunoblot analyses in nuclear extracts and histological findings showing accumulation of Hdac3 and NCoR1 repressors in livers of Vps15-null mice (**new Supplementary Fig. 4c**). The quantification of these immunoblots showed no significant changes in total levels of repressors in livers of Vps15-LKO mice treated with fenofibrate.

Point 8. VPA is a non-specific inhibitor of HDACs and cause many changes in chromatin that are independent of HDAC3/NCoR1 complex. Moreover, the changes of VPA are minimal and never rescue to controls. Authors must perform a genetic approach to specifically block NCoR1 or HDAC3 and show a rescue of the VPS15 phenotype in term of lipid metabolism.

Response:

We agree with the reviewer that the genetic rescue experiment would provide an additional elegant confirmation of Hdac3 repressive role in PPAR α function in Vps15-null models. In this revised manuscript, we include additional data on knockdown of Hdac3 using specific shRNA in Vps15-depleted hepatocytes. First, we show that knockdown of Hdac3 in hepatocytes prepared from AlbCre⁺;Vps15^{fl/fl} mice rescues transcript levels of PPAR α targets such as *Aox* and *Cpt1* (**new Supplementary Fig. 6b, 6c**). Second, we show that defective mitochondrial respiration in Vps15- and Vps34-depleted primary hepatocytes could be significantly increased by acute co-depletion of Hdac3 (**new Fig. 5a** and **new Supplementary Fig. 6a**). Altogether, these novel observations *in vitro* and our findings using VPA *in vivo* support our conclusions on the inhibitory role of Hdac3 in PPAR α driven transcription.

Point 9. Which is the rationale to overexpress VPS15 in presence of PGC1 α ? Is because VPS15overexpression enhances mitophagy? Or just autophagy? In both cases authors must show data (Flux) that support these hypothesis (Suppl Fig 3c is poor and does not support anything). It is also unclear how VPS15 would enhance PPAR α activity (Fig 5d). Is it because NCoR1/HDAC3 are decreased? In this case they must be studied and shown.

Response:

We apologize that we may have been unclear in the description of the figure in previous version. Indeed, the rationale for Vps15 expression was to increase autophagic flux. It is

evidenced by significantly decreased p62 and LC3-II protein levels upon Vps15/PGC1 α expression that we have now quantified (**new Supplementary Fig. 7e**). These are two commonly used read-outs of active autophagy. Autophagy activation is expected consequence of Vps15 expression. It was widely documented in cellular models upon expression of another regulatory subunits of the class 3 PI3K complex, e.g. Beclin protein overexpression(Liang, et al. 1999). Notably, PPAR α transcriptional activity was increased in this experimental setting (**Fig. 6d, 6e**). As suggested by the reviewer, we analysed in the same experiments Hdac3 protein expression and now we show that its protein levels are also decreased upon Vps15/PGC1 α expression (**new Supplementary Fig. 7e**). Following reviewer's suggestion, we have also performed the flux experiment upon Vps15/PGC1 α expression. It shows that in this condition autophagic degradation of Hdac3 is induced (**new Supplementary Fig. 7f**). Altogether, these observations are consistent with the new data presented on **new Fig. 4** showing that Vps15 expression promotes Hdac3 dissociation from PPAR α and increases its binding to Atg8-like proteins.

Point10. EM picture of mitochondria are poor and looks like that cristae are dramatically altered in FENO treated samples. A better characterization and quantification of mito morphology is required.

Response:

We apologize for low resolution of electron microscopy figure in the merged PDF file of submitted manuscript. In this revised version of the manuscript, we have changed the captures of mitochondria in all conditions to higher resolution (**new Fig. 6a**). This figure is now much improved and clearly shows present cristae in hepatocytes of fenofibrate treated mice (**new Fig. 6a**). As suggested by the reviewer, in addition to mitochondria area, we have quantified the mitochondria number in all experimental conditions. Those quantifications revealed that the mitochondrial number is not modified between conditions (**new Fig. 6a**). It is also consistent with our novel data that demonstrates activated fission machinery in livers of Vps15 hepatic mutant (**new Supplementary Fig. 7a**). These additional analyses together with new data on quantification of mitochondrial mass, support our conclusions that while mitochondrial number is not modified in Vps15-null hepatocytes, the mitochondrial mass is significantly reduced upon Vps15 depletion.

Minor points

In the introduction there are many conceptual errors. Firstly, mitochondria are not the major source of phosphatidylethanolamine that is mainly synthesized in the smooth ER. There is only one paper that claimed it but was never confirmed by other studies suggesting that this is peculiar of some cell or experimental setting. Secondly, TFEB or PGC1 α /PPARs play a role in autophagy regulation in liver but this action is not shared with other tissues, certainly not in skeletal/cardiac muscles. Small size mitochondria (lane 99) do not originate as consequence of mitochondrial biogenesis

defect but may also result from excess of fission. The authors should change the world biogenesis with mitochondrial dynamics.

Response:

We apologise for been unclear in the previous version of our manuscript. We have modified the introduction to better convey the message on growing recognition of requirement of functional mitochondria for autophagy. We cited works that demonstrated that mitochondrial respiration, phosphatidylethanolamine production and mitochondrial membrane are required for effective autophagic flux (Hailey, et al. 2010), (Rockenfeller, et al. 2015), (Thomas, et al. 2018). Furthermore, the revised version of the manuscript is now edited to highlight the liver tissue context of reported findings on transcriptional control of autophagy, which is also the focus of our manuscript. As suggested by the reviewer, in addition to PPAR α /PGC1 α dependent mitochondrial gene transcription and biogenesis, we have also mentioned the mechanism of mitochondrial fission as a possible contributor to the observed mitochondrial phenotype of hepatic Vps15 mutant. We thank the reviewer for these insightful comments.

Reviewer #2 (Remarks to the Author):

We would like to thank the reviewer for the interest in our study, for the appreciation of the major novelty regarding the role of class 3 PI3K in control of transcriptional activity of PPAR α and for the constructive criticism that improved our work.

This manuscript reports the study that attempted to demonstrate a class 3 PI3K-mediated mechanism linking autophagy and mitochondrial function to the control of liver lipid metabolism. The presented results showed the transcriptional suppression of liver mitochondrial gene program in liver-specific Vps15 KO mice. Using fenofibrate, Bafilomycin A1 and valproic acid (VPA) treatments as well as PGC-1 α expression, the data indicated that Vps15 might function to regulate PPAR α -dependent mitochondrial metabolism through autophagic degradation of HDAC3/NCoR1. The authors conclude that “the class 3 PI3K acts upstream of nuclear receptors and exerts a broad transcriptional control in the liver to match autophagic activity with mitochondrial metabolism during fasting”.

Overall this is a potentially interesting study that may expand our molecular understanding of the mechanistic links between autophagy, mitochondrial function and lipid homeostasis. However, the current study is kind of descriptive at the gene expression levels; more mechanistic evidence is needed to support that Vps15-dependent autophagy mediated the degradation of HDAC3/NCoR1.

Specific points:

1. Whether Vps15-mediated autophagic degradation of liver HDAC3/NCoR1 can be induced during starvation in WT control mice but not in Vps15-LKO mice remains unclear. The effect of starvation upon the cytoplasmic and nuclear abundances of NCoR1/HDAC3, PPAR α and PGC1 α in the livers of WT versus Vps15-LKO mice can be examined.

Response:

We thank reviewer for raising this important point that we addressed by performing *ex vivo* and *in vivo* flux experiments in liver of control and hepatic mutant of Vps15 (**new Fig. 3e** and **new Supplementary Fig. 4e**). We now show that endogenous NCoR1 and Hdac3 proteins are degraded in lysosomes in control liver unlike in autophagy deficient Vps15 mutant. This degradation is also stimulated by 24-hour fasting (*in vivo* flux) (**new Fig. 3e**) or incubation of liver explants in EBSS media (*ex vivo* flux) (**new Supplementary Fig. 4e**). As suggested by reviewer, we also performed additional immunoblot analyses in total and nuclear liver extracts of fed and fasted for 24 hours control mice and hepatic mutant of Vps15. These new data show that fasting induced nuclear PPAR α accumulation is inefficient in Vps15 mutants and is oppositely correlated with levels of repressor proteins NCoR1 and Hdac3 (**new Supplementary Fig. 4g, 4i**). Altogether, these new data further reinforce our initial findings on the lysosomal degradation of PPAR α repressors.

2. More detailed molecular examination of Vps15-mediated degradation of HDAC3/NCoR1 is desirable in WT and Vps15-LKO hepatocytes (e.g. the co-localization of Vps15/HDAC3 and Vps15/NCoR1 puncta with a lysosome marker such as LAMP2 in response to starvation or other stimuli).

Response:

Following guidance of all three reviewers, in this revised version of manuscript, we present a large amount of new mechanistic data (**see point 7 to Reviewer 1**). Altogether, these data strongly advocate lysosomal degradation of NCoR1 and Hdac3 through selective autophagy by binding to Atg8-like proteins (**new Fig. 4**).

Moreover, following reviewer's suggestion, we also performed additional immunofluorescent analyses in primary hepatocytes in which autophagic flux was blocked by Bafilomycin A1 treatment. In addition to the previously presented co-localization of NCoR1 and Lamp1, in this revised manuscript, we show that both NCoR1 and HDAC3 are targeted to Lamp2-positive lysosomal compartments (**new Fig. 4a**).

3. With regard to the metabolic characteristics, the serum TG levels in Vps15-LKO mice during fasting should also be measured to see if lipid secretion might also be affected.

Response:

We thank the reviewer for asking this interesting question on the role of hepatic class 3 PI3K in whole-body lipid homeostasis. A similar question was raised by the **Reviewer 1**. In response to both comments we have performed the biochemical studies in plasma of Vps15 hepatic mutants in fed and fasted state (**see point 6 to Reviewer 1**). These new data are presented as **new Supplementary Fig. 3f**.

4. Are there changes in mitochondrial mass and number during starvation in WT and Vps15-LKO livers? How would Vsp15 deficiency affect mitophagy?

Response:

We thank the reviewer for raising this important point that led us to a deeper understanding of the role of Vps15 in mitochondrial maintenance. Since similar questions were also raised by the first reviewer, see our response to **point 2 to Reviewer 1 on page 3**. In sum, we have quantified mtDNA content in control and hepatic mutants of Vps15 and PPAR α . We now show, first, that mitochondrial mass is significantly reduced in Vps15 mutant and, second, that fasting induced increase of mtDNA in liver requires Vps15 and PPAR α (**new Fig. 7c, 7d**). Our molecular analyses of mitochondrial fractions also show that pro-mitophagic Parkin pathway is activated in Vps15-null livers. These new data are presented as **new Supplementary Fig. 7a and 7b**.

5. Given the dramatic decrease in nuclear PPARalpha levels in Vps15-LKO livers (Fig. 2f), how would fenofibrate and valproic acid (VPA) treatments or PGC1alpha overexpression exert the observed metabolic effects through PGC1alpha activity? Whether PPARalpha expression levels were changed under these conditions should be examined.

Response:

We thank reviewer for this comment that allowed us to identify the differences in PPAR α protein expression upon pharmacological rescue experiments. In agreement with our previously reported immunohistochemistry in liver tissue samples, PPAR α protein levels were significantly upregulated in livers of fenofibrate treated hepatic Vps15 mutants (**new Supplementary Fig. 4b**). Unlike fenofibrate, treatment with VPA did not have the same normalizing effect on PPAR α protein expression (**new Supplementary Fig. 6d, 6f**). These findings are in agreement with the literature as ligand binding and association with co-activator proteins such as PGC1 α stabilize PPAR α protein. Moreover, since PPAR α is known to control its own transcription, its direct activation with a synthetic ligand such as fenofibrate is expected to promote its expression, unlike an indirect effect of VPA through inhibition of Hdac activity. In VPA treated livers, Hdac activity is reduced, however the potent activation of PPAR α , as observed in fenofibrate treated mice, is not achieved. Therefore, we believe the ligand availability (e.g. free fatty acids that we find depleted in livers of Vps15 mutants) could be one of the causes as to why VPA treatment is less effective as compared to fenofibrate *in vivo*.

6. In Fig 1d, what is the identity of the lower protein band detected upon CRE-mediated depletion of Vps15? The protein sizes need to be indicated for all the immunoblots.

Response:

We apologize for the lack of clarity of our previous version of the manuscript. The lower migrating protein band on immunoblot with anti-Vps15 antibody is a truncated non-functional form of Vps15 that is transcribed from the start codon in exon 4 upon effective Cre-mediated recombination in Vps15 gene locus. We have characterized this truncated protein in our previous report and demonstrated that it has no dominant negative effect (Nemazanyy, EMBO Mol.Med, 2013). We have included this information in the text of revised manuscript. We have also indicated the protein sizes on all immunoblots.

7. Fig 1a showed the downregulated genes. What about the upregulated genes? Do they (1466 upregulated genes?) also contribute to the liver phenotypes?

Response:

We thank the reviewer for this comment that improved the presentation of our findings. We have now provided bioinformatic analyses on significantly upregulated genes (David pathway analysis). These new analyses are presented as **new Supplementary Fig.1a** and **new Supplementary Data 2**. The complete list of up- and down-regulated genes is also presented in **Supplementary Data 1**.

8. Mitochondrial OCR analysis can also be done in Vps15-LKO versus WT liver samples.

Response:

This important question on the mitochondrial respiration in livers of Vps15 mutant was raised by three reviewers. We addressed it, see detailed response to **point 2** of **Reviewer 1**. Briefly, we measured activity of mitochondrial respiratory chain complexes in liver extracts of control and Vps15 liver-specific knockout mice. Those biochemical analyses showed decreased respiratory chain activity in livers of Vps15 mutants. These findings are presented as **new Fig. 1d** and **new Supplementary Fig. 2c**.

Reviewer #3 (Remarks to the Author):

We would like to thank the reviewer for the appreciation of our work and for noting its integrative aspect in use of different experimental approaches and the importance of our findings for the field. We are also grateful for the reviewer's comments that allowed us significantly improve our work.

This manuscript by Iershov and colleagues address the role of Class III PI3K in liver adaptation to fasting through genetic depletion of the Vps15 regulatory subunit. In previous work they had characterized the metabolic reprogramming occurring in hepatocytes upon Vps15 inactivation. Here they profile these changes by microarray and metabolomic analysis finding that both mitochondrial functions and fatty acid metabolism are significantly downregulated in Vps15^{-/-} hepatocytes. This is linked to a reduction in PPAR α and PGC1 α (through an unknown mechanism) and an accumulation in NCoR/HDAC3 (through impaired lysosomal degradation). The manuscript is well written and the findings are interesting as they reveal a novel role for the Class III PI3K, or at least the Vps15 subunit, in regulating the transcriptional response to fasting through PPAR α and cofactors even though it is not completely clear whether some of these effects might be regulated through insulin signalling (which was reported upregulated in the same genetic model) rather than by direct modulation of the PPAR/PGC1 axis. Overall, these studies are relevant to the field as they improve our understanding of adaptive mechanisms. However, areas that should be addressed/improved prior to publication include:

Major comments

1. Is this specific to deletion of Vps15, which might have unknown functions outside of the PI3K complex, or can be recapitulated by manipulation of the Vps34 subunit? The authors have previously reported (Jaber, PNAS 2012) Vps34 genetic deletion and describe its liver phenotype, so the comparison should be doable.

Response:

We thank reviewer for raising this interesting point that led us to strengthen the control of PPAR α transcriptional activity by the autophagic degradation of its repressors. We have addressed this question by using in selected assays the pharmacologic inhibitor of Vps34 lipid kinase, PIK-III, and by knocking down Vps34 using specific shRNA. In both approaches, we have observed that targeting Vps34 expression and activity was sufficient to inhibit the mitochondrial respiration and recruitment of NCoR1 and Hdac3 to Atg8-like proteins. These new data are presented as **new Fig. 4b** and **new Supplementary Fig. 6a**. In addition, our comparative analyses of publicly available microarray from autophagy deficient Atg7 liver knockout mice show significant overlap in transcriptional responses with PPAR α and Vps15 hepatic mutants (**Figure for Reviewer 6**). Although we cannot exclude that the Vps15 protein might have a role beyond the complex with lipid kinase Vps34, our analyses point to the mechanism of selective autophagy of transcriptional repressors dependent on lipid kinase activity of Vps34/Vps15 complex.

Figure for Reviewer 6. Transcriptional responses are similar in Atg7, Vps15 and PPAR α LKO mice. Heat map representing data from a gene expression profiling experiments performed with liver samples of mice of indicated genotypes (GSE35015 for PPAR α ^{-/-}, GSE73299 for AlbCre⁺;PPAR α ^{ff}, GSE65174 for AlbCre⁺;Atg7^{ff}). The genes are grouped according to their biological function revealed by GO annotation. The gene names corresponding to GO groups are listed in Supplementary Data 4.

2. The mitochondrial phenotype needs further characterization. As presented, it is unclear whether the defect is in mitochondrial number, because of impaired mitochondrial biogenesis, or mitochondrial size/function. For example, the decrease in cytc staining (Fig.1c) cannot be used as a measure of mitochondria depletion as it may be its expression being downregulated. Similarly, mitochondrial area should not be considered an equivalent to mitochondrial mass (Fig 5a). In fact, a question that is not fully address is whether mitochondria are smaller or reduced in number or both? This is a critical question for a correct interpretation of the Seahorse results (Fig.1d).

Response:

We thank reviewer for raising these important questions that allowed us better understand the impact of class 3 PI3K inactivation on mitochondrial function. Since similar questions were also raised by the other reviewers, see our **point 2 to the Reviewer 1**. Briefly, we have performed additional immunofluorescent analyses with Tom40 as a mitochondrial membrane protein (**new Fig. 1c**), we have quantified the mitochondrial number on the electron microscopy captures (**new Fig. 6a**), we have measured the mitochondrial mass (mtDNA) (**new Fig. 7c, 7d, new Supplementary Fig. 2b**) and mitochondrial respiration activity (**new Fig. 1d, new Supplementary Fig. 2c**) in liver tissue of Vps15-null mice. These new results together with the findings already presented in the previous version allow us to

conclude that class 3 PI3K acts upstream of mitochondrial biogenesis and mitochondrial maintenance in hepatocytes.

3. It is unclear how the accumulation of corepressors can influence PPARalpha-mediated transcription if PPARalpha is not present (nuclear expression is completely absent in Fig.2F). If the authors think there may be residual levels that are sufficient for recruiting the corepressor to PPARalpha target genes, this should be confirmed by chromatin immunoprecipitation. An alternative possibility is that accumulation of NCoR and HDAC3 affects transcription through other nuclear receptors.

Response:

We apologize for being unclear on this important point. Our data suggest that although protein levels of PPAR α is drastically decreased, it is still expressed at low levels in Vps15-null livers. This is evidenced by the new immunoblot analyses presented in **new Supplementary Fig. 4b, 4g, 6d and 6f** and could be also seen on immunohistological analyses presented in the previous version (**Fig. 3c**). This conclusion is fully consistent with the observations that PPAR α transcriptional activity could be partially restored either by fenofibrate or by VPA treatment. Although we cannot exclude that other transcription factors might contribute to the phenotype of hepatic mutants of Vps15, the rescue that was achieved with fenofibrate, a selective ligand of PPAR α , and lack of responses in PPAR α hepatic mutant suggest that the mitochondrial dysfunction in Vps15-null hepatocytes is largely PPAR α -dependent. We have modified the text of the revised manuscript to convey this point on residual PPAR α expression.

4. What is the rationale for the luciferase experiment in Fig5d? Obviously overexpression of a PPAR coactivator will activate a PPARE reporter, is there any additional conclusion to be made in the context of these studies? Also it is unclear to which extent overexpression of PGC1a is sufficient to rescue for the lack of Vsp15 or whether this is restricted to the regulation of mitochondrial gene expression, known to be regulated by PGC1alpha. This should be clarified in the final model if necessary.

Response:

We apologize that we may have been unclear in our description. In this experiment, we asked whether co-expression of Vps15 would potentiate a co-activating effect of PGC1 α , which indeed was the case (**Fig. 6d and 6e**). We have now provided the evidence that expression of Vps15 in these conditions promoted autophagic flux and degradation of Hdac3 protein (**new Supplementary Fig. 7f**). In this revised manuscript, we also show that overexpression of Vps15 is sufficient to promote an association of Hdac3 and NCoR1 co-repressors with Atg8-like proteins (**new Fig. 4c**). However, we cannot exclude and will test in future that Vps15 protein might have a wider and positive role in the control of PGC1 α stability and transcription co-activating function. Finally, following the reviewer suggestion we also measured in this experimental setting other PPAR α target involved in fatty acid degradation

and now show that expression of *Cpt1*, a carnitine fatty acid transporter, is increased by PGC1 α expression in Vps15-depleted cells (**new Fig. 6f**). These findings are consistent with our gain-of-function experiments presented in the previous version of the manuscript (**Fig.6e**).

Minor comments

1. What are the pathways enriched among upregulated genes in Fig 1a? The DEG are all listed in the Data Table, but there is no comment on what are the upregulated genes (>50% of total). The entire program should be discussed even if the focus is later on the downregulated program.

Response:

We thank the reviewer for this comment that significantly improved the presentation of our findings. In response to this question, that was also raised by other reviewers, we have now provided the bioinformatic analyses on significantly upregulated genes (David pathway analysis). These new analyses are discussed in the text and presented as **new Supplementary Fig. 1a** and **new Supplementary Data 2**.

2. What happens to adipose tissue lipolysis in these mice?

Response:

We thank the reviewer for raising this important question that was also asked by other reviewers. We have addressed it by performing DEXA analyses and biochemical analyses of plasma in fasted and fed control and Vps15 hepatic mutants which are now presented as **new Supplementary Fig. 3** (detailed response in **point 3/6 to the Reviewer 1**

3. In describing NCoR/HDAC3 regulation the authors describe it, in multiple instances, as “nuclear exclusion”. It is unclear what are the basis for this conclusion as there is no sign of them being regulated through changes in intracellular localization.

Response:

We thank the reviewer for pointing to possible misunderstanding in our formulations, indeed we meant “levels” and have modified the text accordingly.

4. Fig3e: the increase in HDAC3 protein level should be confirmed by quantification of the blot. The increase in NCoR levels is clear, but HDAC3 change is minimal and Actin itself appears increased to the same extent.

Response:

As requested by the reviewer, we have added a quantification of the flux experiment in hepatocytes (**new Fig. 3f**). In addition, in response to this comment of the reviewer and comments of other reviewers on the autophagic flux of these transcriptional repressors, we have now performed *in vivo* and *ex vivo* flux analyses in liver of control and Vps15 mutants (**new Fig. 3e** and **new Supplementary Fig. 4e**). The lysosomal degradation of NCoR1 and Hdac3 proteins is also supported by our new immunofluorescent analyses (**new Fig. 4a**). This

conclusion is further reinforced by our novel findings of interaction between NCoR1 and Hdac3 and Atg8-like proteins (**new Fig. 4b, 4c**).

5. To our knowledge VPA is not an HDAC3 specific inhibitor, thus the results of VPA-mediated rescue should not be interpreted as a confirmation of HDAC3 role in this regulatory axis, other HDACs might be involved.

Response:

We apologize for being unclear in the previous version of the manuscript. We have now modified the text to highlight that other Hdacs might contribute to the phenotype of Vps15 hepatic mutants. To this end, we now show that levels of Hdac1 are also increased in livers of Vps15 mutants (**new Supplementary Fig. 6d**). Furthermore, in this revised version of the manuscript, we show that acute knockdown of Hdac3 partially restores mitochondrial activity and expression of PPAR α targets in Vps15-null hepatocytes (**new Fig.5a** and **new Supplementary Fig. 6a**). These new data together with evidence already presented in the previous version advocate the role of the repressors in PPAR α inhibition in liver of Vps15 mutant mice.

6. Representation of the microarray results with black background is hard to see. If possible, changing color scheme would help readers.

Response:

We thank reviewer for this insightful suggestion that has improved the presentation of our data. As suggested by the reviewer, we have converted the colour code in the heatmap of microarray (**new Fig. 2d** and **7a**).

Bekesi, A., and D. H. Williamson

1990 An explanation for ketogenesis by the intestine of the suckling rat: the presence of an active hydroxymethylglutaryl-coenzyme A pathway. *Biol Neonate* 58(3):160-5.

Hailey, D. W., et al.

2010 Mitochondria supply membranes for autophagosome biogenesis during starvation. *Cell* 141(4):656-67.

Leone, T. C., C. J. Weinheimer, and D. P. Kelly

1999 A critical role for the peroxisome proliferator-activated receptor alpha (PPARalpha) in the cellular fasting response: the PPARalpha-null mouse as a model of fatty acid oxidation disorders. *Proc Natl Acad Sci U S A* 96(13):7473-8.

Liang, X. H., et al.

1999 Induction of autophagy and inhibition of tumorigenesis by beclin 1. *Nature* 402(6762):672-6.

Montagner, A., et al.

2016 Liver PPARalpha is crucial for whole-body fatty acid homeostasis and is protective against NAFLD. *Gut* 65(7):1202-14.

Nemazanyy, I., et al.

2015 Class III PI3K regulates organismal glucose homeostasis by providing negative feedback on hepatic insulin signalling. *Nat Commun* 6:8283.

Owen, O. E., et al.

1969 Liver and kidney metabolism during prolonged starvation. *J Clin Invest* 48(3):574-83.

Penhoat, A., et al.

2014 Intestinal gluconeogenesis is crucial to maintain a physiological fasting glycemia in the absence of hepatic glucose production in mice. *Metabolism* 63(1):104-11.

Rockenfeller, P., et al.

2015 Phosphatidylethanolamine positively regulates autophagy and longevity. *Cell Death Differ* 22(3):499-508.

Takagi, A., et al.

2016 Mammalian autophagy is essential for hepatic and renal ketogenesis during starvation. *Sci Rep* 6:18944.

Thomas, H. E., et al.

2018 Mitochondrial Complex I Activity Is Required for Maximal Autophagy. *Cell Rep* 24(9):2404-2417 e8.

Zhang, D., et al.

2011 Proteomics analysis reveals diabetic kidney as a ketogenic organ in type 2 diabetes. *Am J Physiol Endocrinol Metab* 300(2):E287-95.

Zhao, Z., et al.

2018 Hepatic PPARalpha function is controlled by polyubiquitination and proteasome-mediated degradation through the coordinated actions of PAQR3 and HUWE1. *Hepatology* 68(1):289-303.